# RAG-DDR: Optimizing Retrieval-Augmented Generation Using Differentiable Data Rewards

**Xinze Li**[1*] **Sen Mei**[1*] **Zhenghao Liu**[1†] **Yukun Yan**[2†] **Shuo Wang**[2] **Shi Yu**[2]
**Zheni Zeng**[2] **Hao Chen**[2] **Ge Yu**[1] **Zhiyuan Liu**[2] **Maosong Sun**[2] **Chenyan Xiong**[3]
[1]Northeastern University   [2]Tsinghua University   [3]Carnegie Mellon University

## Abstract

Retrieval-Augmented Generation (RAG) has proven its effectiveness in mitigating hallucinations in Large Language Models (LLMs) by retrieving knowledge from external resources. To adapt LLMs for the RAG systems, current approaches use instruction tuning to optimize LLMs, improving their ability to utilize retrieved knowledge. This supervised fine-tuning (SFT) approach focuses on equipping LLMs to handle diverse RAG tasks using different instructions. However, it trains RAG modules to overfit training signals and overlooks the varying data preferences among agents within the RAG system. In this paper, we propose a Differentiable Data Rewards (DDR) method, which end-to-end trains RAG systems by aligning data preferences between different RAG modules. DDR works by collecting the rewards to optimize each agent in the RAG system with the rollout method, which prompts agents to sample some potential responses as perturbations, evaluates the impact of these perturbations on the whole RAG system, and subsequently optimizes the agent to produce outputs that improve the performance of the RAG system. Our experiments on various knowledge-intensive tasks demonstrate that DDR significantly outperforms the SFT method, particularly for LLMs with smaller-scale parameters that depend more on the retrieved knowledge. Additionally, DDR exhibits a stronger capability to align the data preference between RAG modules. The DDR method makes the generation module more effective in extracting key information from documents and mitigating conflicts between parametric memory and external knowledge. All codes are available at `https://github.com/OpenMatch/RAG-DDR`.

## 1 Introduction

Large Language Models (LLMs) have demonstrated impressive capabilities in language understanding, reasoning, and planning across a wide range of natural language processing (NLP) tasks (Achiam et al., 2023; Touvron et al., 2023; Hu et al., 2024). However, LLMs usually generate incorrect responses due to hallucination (Ji et al., 2023; Xu et al., 2024c). To alleviate this problem, existing studies employ Retrieval-Augmented Generation (RAG) (Lewis et al., 2020; Shi et al., 2023; Peng et al., 2023) to enhance the ability of LLMs and help LLMs access long-tailed knowledge and up-to-date knowledge from different data sources (Trivedi et al., 2023; He et al., 2021; Cai et al., 2019; Parvez et al., 2021). However, the conflict between retrieved knowledge and parametric memory usually misleads LLMs, challenging the effectiveness of the RAG system (Li et al., 2022; Chen et al., 2023; Asai et al., 2024).

To ensure the effectiveness of RAG systems, existing research has focused on developing various agents to obtain high-quality external knowledge (Gao et al., 2024; Xu et al., 2024a; Jiang et al., 2023; Xu et al., 2024b), which can refine retrieval documents through query reformulation, reranking candidate documents, summarizing the retrieved documents, or performing additional actions to obtain more relevant information for LLMs (Yan et al., 2024; Trivedi et al., 2023; Asai et al., 2023;

---

[*] Equal Contribution.
[†] Corresponding Authors.

Yu et al., 2023a). To further optimize the RAG system, some methods independently optimize different RAG modules by using the EM method (Singh et al., 2021; Sachan et al., 2021) or build the instruction tuning dataset for Supervised Fine-Tuning (SFT) these LLMs-based RAG modules (Lin et al., 2023; Asai et al., 2023). However, these SFT-based methods usually make LLMs overfit the training signals and face the catastrophic forgetting problem (Luo et al., 2023b).

Furthermore, current research aims to optimize RAG modules by aligning their data preferences and primarily focuses on optimizing a two-agent RAG system, which consists of retrieval and generation modules. Typically, these methods train only the retrieval module by using the preference signals from the generation module, making the retrieval module supply more accurate documents to satisfy the data preference of the generation module (Shi et al., 2023; Yu et al., 2023b). However, these methods only optimize the retrieval module and overlook that the generation module still faces the knowledge conflict (Xie et al., 2024), making the generation module's outputs do not align with the data preferences of the RAG system. Thus, optimizing and aligning the data preferences of each module in the RAG system is essential for building a more tailored RAG system.

This paper introduces a Differentiable Data Rewards (DDR) method for end-to-end optimizing each agent in the RAG system using the DPO (Rafailov et al., 2024) method. DDR uses a roll-out method (Kocsis & Szepesvári, 2006) to collect the reward from the overall system for each agent and optimizes the agent according to the reward. Specifically, we follow Asai et al. (2023) and build a typical RAG system to evaluate the effectiveness of DDR model. This RAG system consists of a knowledge refinement module for selecting retrieved documents and a generation module for producing responses based on the query and refined knowledge. Then we conduct the RAG-DDR model by optimizing the two-agent based RAG system using DDR. We use the reward from the entire RAG system and iteratively optimize both the generation and knowledge refinement modules to align data preferences across both agents, enabling the RAG system to generate accurate responses.

Our experiments on various Large Language Models (LLMs) demonstrate that Differentiable Data Rewards (DDR) outperforms all baseline models, achieving significant improvements over the previous method (Lin et al., 2023) in a range of knowledge-intensive tasks. DDR can effectively retrofit LLMs for the RAG modeling and help LLMs generate higher-quality responses of an appropriate length. Our further analyses show that the effectiveness of our RAG-DDR model primarily derives from the generation module, which is optimized by the reward from the RAG system. The DDR-optimized generation module is more effective in capturing crucial information from retrieved documents and alleviating the knowledge conflict between external knowledge and parametric memory. Further analyses show that the effectiveness of DDR-optimized RAG systems can be generalized even when additional noisy documents are incorporated during response generation.

## 2 RELATED WORK

Retrieval-Augmented Generation (RAG) is widely used in various real-world tasks, such as open-domain QA (Trivedi et al., 2023), language modeling (He et al., 2021), dialogue (Cai et al., 2019), and code generation (Parvez et al., 2021). RAG models retrieve documents from external corpus (Karpukhin et al., 2020; Xiong et al., 2021) and then augment the LLM's generation by incorporating documents as the context of input (Ram et al., 2023) or aggregating the output probabilities from the encoding pass of each retrieved document (Shi et al., 2023). They help LLMs alleviate hallucinations and generate more accurate and trustworthy responses (Jiang et al., 2023; Xu et al., 2023; Luo et al., 2023a; Hu et al., 2023; Kandpal et al., 2023). However, retrieved documents inevitably incorporate noisy information, limiting the effectiveness of RAG systems in generating accurate responses (Xu et al., 2024a;b; Longpre et al., 2021; Liu et al., 2024b).

Some studies have demonstrated that the noise from retrieved documents can mislead LLMs, sometimes resulting in degraded performance even on some knowledge-intensive tasks (Foulds et al., 2024; Shuster et al., 2021; Xu et al., 2024b). This issue primarily derives from the knowledge conflict between the parametric knowledge of LLMs and external knowledge (Jin et al., 2024; Longpre et al., 2021; Chen et al., 2022; Xie et al., 2024; Wu et al., 2024). Xie et al. (2024) have demonstrated that LLMs are highly receptive to external evidence when external knowledge conflicts with the parametric memory (Xie et al., 2024). Thus, lots of RAG models focus on building modular RAG pipelines to improve the quality of retrieved documents (Gao et al., 2024). Most of them aim to conduct more accurate retrieval models by employing a retrieval evaluator to trigger different

knowledge refinement actions (Yan et al., 2024), prompting LLMs to summarize the query-related knowledge from retrieved documents (Yu et al., 2023a) or training LLMs to learn how to retrieve and utilize knowledge on-demand by self-reflection (Asai et al., 2023).

Optimizing the RAG system is a crucial research direction to help LLMs generate more accurate responses. Previous work builds a RAG system based on pretrained language models and conducts an end-to-end training method (Singh et al., 2021; Sachan et al., 2021). They regard retrieval decisions as latent variables and then iteratively optimize the retriever and generator to fit the golden answers. Recent research primarily focuses on optimizing the LLMs in RAG systems. INFO-RAG (Xu et al., 2024a) focuses on enabling LLMs with the in-context denoising ability by designing an unsupervised pretraining method to teach LLMs to refine information from retrieved contexts. RA-DIT (Lin et al., 2023) builds a supervised training dataset and then optimizes the retriever and LLM by instruction tuning. However, these training methods focus on training LLMs to fit the training signals and face the issue of catastrophic forgetting during instruction tuning (Luo et al., 2023b).

Reinforcement Learning (RL) algorithms (Schulman et al., 2017), such as Direct Preference Optimization (DPO) (Rafailov et al., 2024), are widely used to optimize LLMs for aligning with human preferences and enhancing the consistency of generated responses (Putta et al., 2024). Agent Q integrates MCTS and DPO to allow agents to learn from both successful and unsuccessful trajectories, thereby improving their performance in complex reasoning tasks (Putta et al., 2024). STEP-DPO further considers optimizing each inference step of a complex task as the fundamental unit for preference learning, which enhances the long-chain reasoning capabilities of LLMs (Lai et al., 2024). While these models primarily target the optimization of individual agents to improve response accuracy at each step, they do not focus on the effectiveness of data alignment within the multi-agent system. Instead of using SFT methods for RAG optimization (Lin et al., 2023), this paper focuses on using the DPO method to avoid overfitting the training signals and aligning data preferences across different agents, which is different from the above RL-based optimization methods.

## 3 RAG TRAINING WITH DIFFERENTIABLE DATA REWARDS

This section introduces the Differentiable Data Rewards (DDR) method. We first introduce the DDR method, which optimizes the RAG system by aligning the data preferences between different agents (Sec. 3.1). Then, we employ knowledge refinement and generation modules to build the RAG system and utilize the DDR method to optimize the agents in this RAG system (Sec. 3.2).

### 3.1 DATA PREFERENCE LEARNING WITH DIFFERENTIABLE DATA REWARDS

In a RAG system $\mathcal{V} = \{V_1, \ldots, V_t, \ldots, V_T\}$, agents exchange and communicate data. To optimize this system, we first forward-propagate data among agents and then evaluate the performance of the RAG system. Then we backward-propagate rewards to refine the data preferences of each agent.

**Data Propagation.** During communication, the $t$-th agent, $V_t$, acts as both a sender and a receiver. Agent $V_t$ receives data from agent $V_{t-1}$ and simultaneously passes data to agent $V_{t+1}$:

$$x \rightsquigarrow V_1 \ldots V_t \xrightarrow{y_t} V_{t+1} \ldots V_{T-1} \xrightarrow{y_{T-1}} V_T \rightsquigarrow y_T, \tag{1}$$

where $V_t \xrightarrow{y_t} V_{t+1}$ denotes that the agent generates one response $y_t$ of the maximum prediction probability and sends it to the agent $V_{t+1}$. $x \rightsquigarrow$ and $\rightsquigarrow y_T$ represent sending the input $x$ to the RAG system $\mathcal{V}$ and getting the final output $y_T$ from $\mathcal{V}$. The performance of the RAG system $\mathcal{V}$ can be evaluated by calculating the quality score $S(y_T)$ of the final output $y_T$.

**Differentiable Data Reward.** Unlike conventional supervised fine-tuning approaches (Lin et al., 2023), DDR tries to optimize the agent $V_t$ to align its data preference with $V_{t+1:T}$, making the RAG system produce a better response with a higher evaluation score $S(y_T)$.

To optimize the agent $V_t$, DDR aims to propagate the system reward to train the targeted agent. Specifically, we first instruct $V_t$ to sample multiple outputs $\tilde{y}_t$, which incorporate some perturbations into the RAG system. Then, we calculate the reward $r(x, \tilde{y}_t)$ with a rollout process (Kocsis & Szepesvári, 2006). In detail, we regard the agents $\mathcal{V}_{t+1:T}$ as the evaluation model, feed $\tilde{y}_t$ to this subsystem $\mathcal{V}_{t+1:T}$ and calculate the evaluation score $S(y_T)$ of the final output $y_T$:

$$\tilde{y}_t \rightsquigarrow V_{t+1} \xrightarrow{y_t} V_{t+2} \ldots V_{T-1} \xrightarrow{y_{T-1}} V_T \rightsquigarrow y_T, \quad r(x, \tilde{y}_t) = S(y_T). \tag{2}$$

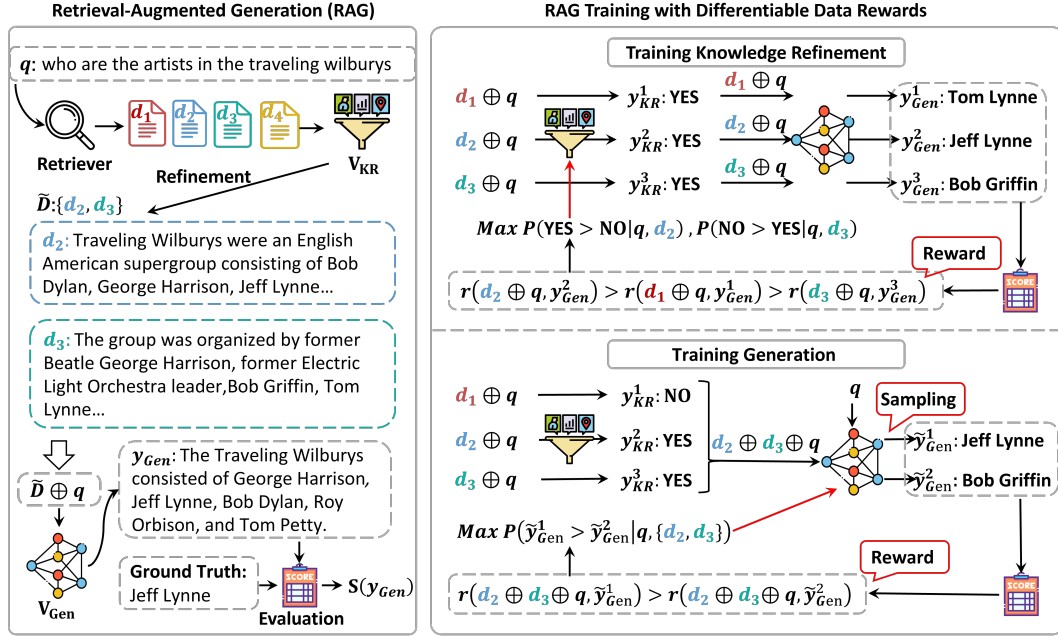

Figure 1: The Illustration of End-to-End Retrieval-Augmented Generation (RAG) Training with Our Differentiable Data Reward (DDR) Method. During training, we iteratively optimize the Generation module ($V_{\text{Gen}}$) and Knowledge Refinement module ($V_{\text{KR}}$).

Finally, we maximize the probability of generating $\tilde{y}_t^+$ over $\tilde{y}_t^-$, where $\tilde{y}_t^+$ wins higher reward than $\tilde{y}_t^-$ $(r(x, \tilde{y}_t^+) > r(x, \tilde{y}_t^-))$:

$$P(\tilde{y}_t^+ > \tilde{y}_t^- | x) = \sigma(r(x, \tilde{y}_t^+) - r(x, \tilde{y}_t^-)), \tag{3}$$

where $\sigma$ is the Sigmoid function. The parameters of $V_t$ can be trained using the DPO (Rafailov et al., 2024) training loss, aiding the DDR model in identifying optimization directions by contrastively learning from the positive ($\tilde{y}_t^+$) and negative ($\tilde{y}_t^-$) outputs:

$$\mathcal{L}(V_t; V_t^{\text{ref}}) = -\mathbb{E}_{(x,\tilde{y}_t^+,\tilde{y}_t^-)\sim\mathcal{D}}[\log \sigma(\beta \log \frac{V_t(\tilde{y}_t^+ \mid x)}{V_t^{\text{ref}}(\tilde{y}_t^+ \mid x)} - \beta \log \frac{V_t(\tilde{y}_t^- \mid x)}{V_t^{\text{ref}}(\tilde{y}_t^- \mid x)})], \tag{4}$$

where $\beta$ is a hyperparameter. $\mathcal{D}$ is the dataset containing the input $x$ and its corresponding preference data pairs $(\tilde{y}_t^+, \tilde{y}_t^-)$. $V_t^{\text{ref}}$ is the reference model, which is frozen during training.

## 3.2 OPTIMIZING A SPECIFIC RAG SYSTEM THROUGH DDR

Given a query $q$ and a set of retrieved documents $D = \{d_1, \ldots, d_n\}$, we build a RAG system by employing a knowledge refinement module ($V_{\text{KR}}$) to filter unrelated documents and a generation module ($V_{\text{Gen}}$) to answer the question. These two modules can be represented by a two-agent system:

$$\{q, D\} \rightsquigarrow V_{\text{KR}} \xrightarrow{\{q,\tilde{D}\}} V_{\text{Gen}} \rightsquigarrow y_{\text{Gen}}, \tag{5}$$

where $\tilde{D} \subseteq D$ and $V_{\text{KR}}$ produces the select actions to filter out the noise documents in $D$ to build $\tilde{D}$. $V_{\text{Gen}}$ generates the answer according to the query $q$ with filtered documents $\tilde{D}$. As shown in Figure 1, we iteratively tune different modules to conduct the RAG-DDR model, beginning with the generation module optimization and subsequently focusing on tuning the knowledge refinement module during training this RAG system. In the rest of this subsection, we will explain the details of how to optimize $V_{\text{KR}}$ and $V_{\text{Gen}}$ using DDR.

**Knowledge Refinement Module.** We follow Asai et al. (2023) and build the knowledge refinement module $V_{\text{KR}}$ to estimate the relevance of each document $d_i$ to the query $q$ for refinement. We feed

both query $q$ and document $d_i$ to the $V_{KR}$ and ask it to produce the action $y_{KR}^i \in \{\text{"YES"}, \text{"NO"}\}$, which indicates whether $d_i$ is retained ($y_{KR}^i = \text{"YES"}$) or discarded ($y_{KR}^i = \text{"NO"}$):

$$y_{KR}^i = \text{LLM}(\text{Instruct}_{KR}, d_i \oplus q), \tag{6}$$

where $\oplus$ denotes the concatenation operation, and $\text{Instruct}_{KR}$ is a prompt designed for knowledge refinement. Then the refined document collection $\tilde{D} = \{d_1, \ldots, d_k\}$ is constructed, where $k <= n$. The document $d_i$ in $\tilde{D}$ leading the RAG system to achieve the highest evaluation reward $r(x, y_{KR}^i = \text{"YES"})$ is considered positive, while the document $d_j$ that results in the lowest reward $r(x, y_{KR}^j = \text{"YES"})$ is regarded as negative. $V_{KR}$ is trained to maximize the probability $P(y_{KR}^i = \text{"YES"} > y_{KR}^i = \text{"NO"}|q, d_i)$ to retain the positive document and maximize the probability $P(y_{KR}^j = \text{"NO"} > y_{KR}^j = \text{"YES"}|q, d_j)$ to filter out irrelevant documents.

**Generation Module.** After knowledge refinement, the query $q$ and filtered documents $\tilde{D} = \{d_1, \ldots, d_k\}$ are fed to the generation module $V_{Gen}$. The response $\tilde{y}_{Gen}$ is sampled from $V_{Gen}$:

$$\tilde{y}_{Gen} \sim \text{LLM}(\text{Instruct}_{Gen}, d_1 \oplus \cdots \oplus d_k \oplus q), \tag{7}$$

where $\text{Instruct}_{Gen}$ is a prompt for generating a tailored response. To reduce the misleading knowledge from the retrieved documents, we also sample responses using only the query as input:

$$\tilde{y}_{Gen} \sim \text{LLM}(\text{Instruct}_{Gen}, q). \tag{8}$$

The response that achieves the highest evaluation score $S(\tilde{y}_{Gen})$ is considered positive ($\tilde{y}_{Gen}^+$), while the lowest evaluation score is considered negative ($\tilde{y}_{Gen}^-$). The generation module $V_{Gen}$ is optimized to maximize the probability of generating the positive response $P(\tilde{y}_{Gen}^+ > \tilde{y}_{Gen}^-|q, \tilde{D})$ to win a higher reward. By generating responses $\tilde{y}_{Gen}$ based on documents or only the query, LLMs can learn to balance internal and external knowledge, alleviating the knowledge conflict problem.

## 4 EXPERIMENTAL METHODOLOGY

This section first describes datasets, evaluation metrics, and baselines. Then, we introduce the implementation details of our experiments. More experimental details are shown in Appendix A.1.

**Dataset.** In our experiments, we follow RA-DIT (Lin et al., 2023) and use the instruction tuning datasets for training and evaluating RAG models. For all datasets and all baselines, we use bge-large (Xiao et al., 2023) to retrieve documents from the MS MARCO 2.1 (Bajaj et al., 2016).

During the training of DDR, we collect ten datasets covering two tasks, open-domain QA and reasoning. Specifically, we randomly sample 32,805 samples for the training set and 2,000 samples for the development set in our experiments. Following previous work (Lin et al., 2023; Xu et al., 2024a), we select the knowledge-intensive tasks for evaluation, including open-domain question answering, multi-hop question answering, slot filling, and dialogue tasks. The open-domain QA tasks consist of NQ (Kwiatkowski et al., 2019), MARCO QA (Bajaj et al., 2016) and TriviaQA (Joshi et al., 2017), which require models to retrieve factual knowledge to help LLMs answer the given question. For more complex tasks, such as multi-hop QA and dialogue, we use HotpotQA dataset (Yang et al., 2018) and Wikipedia of Wizard (WoW) (Dinan et al., 2019) for evaluation. Besides, we also employ T-REx (Elsahar et al., 2018) to measure one-hop fact look-up abilities of models.

**Evaluation.** Following Xu et al. (2024a), we utilize Rouge-L and F1 as evaluation metrics for the MARCO QA task and WoW task, respectively. For the rest tasks, we use Accuracy.

**Baselines.** In our experiments, we compare DDR with five baseline models, including zero-shot models and supervised finetuning models. We first treat the LLM as a black box and conduct three baselines, including LLM w/o RAG, Vanilla RAG and REPLUG. For the LLM w/o RAG model, we directly feed the query to the LLM and ask it to produce the answer according to its memorized knowledge. To implement the vanilla RAG model, we follow previous work (Lin et al., 2023), use retrieved documents as context and leverage the in-context learning method to conduct the RAG modeling. REPLUG (Shi et al., 2023) is compared, which ensembles output probabilities from different passage channels. Self-RAG (Asai et al., 2023) is employed as a baseline, which trains the LLMs to retrieve documents on-demand and reflect on the retrieved documents, and generate responses using a reflection token. RA-DIT (Lin et al., 2023) is also compared in our experiments,

Table 1: Overall Performance of Different RAG Models. The **best** and second best results are highlighted. In our experiments, we employ Llama3-8B as the knowledge refinement module and utilize LLMs of varying scales (Llama3-8B and MiniCPM-2.4B) as the generation module.

| Method | Open-Domain QA | | | Multi-Hop QA | Slot Filling | Dialogue |
|---|---|---|---|---|---|---|
| | NQ | TriviaQA | MARCO QA | HotpotQA | T-REx | WoW |
| *MiniCPM-2.4B* | | | | | | |
| LLM w/o RAG | 20.1 | 45.0 | 17.1 | 17.7 | 22.6 | 14.9 |
| Vanilla RAG (2023) | 42.2 | 79.5 | 16.7 | 26.7 | 22.1 | 14.4 |
| REPLUG (2023) | 39.4 | 77.0 | 19.4 | 24.7 | 26.7 | 15.7 |
| Self-RAG (2023) | 28.8 | 48.6 | 18.2 | 15.2 | 23.9 | 13.4 |
| RA-DIT (2023) | 41.8 | 78.6 | 19.6 | 26.1 | 25.2 | 15.7 |
| RAG-DDR (w/ 1-Round) | 47.0 | 82.7 | 28.1 | 32.5 | **32.6** | 17.2 |
| RAG-DDR (w/ 2-Round) | **47.6** | **83.6** | **29.8** | **33.2** | 32.1 | **18.2** |
| *Llama3-8B* | | | | | | |
| LLM w/o RAG | 35.4 | 78.4 | 17.0 | 27.7 | 16.0 | 14.6 |
| Vanilla RAG (2023) | 46.2 | 84.0 | 20.6 | 30.1 | 26.9 | 12.5 |
| REPLUG (2023) | 45.1 | 83.1 | 22.5 | 29.4 | 23.1 | 14.0 |
| Self-RAG (2023) | 39.6 | 78.2 | 12.5 | 24.3 | 29.5 | 14.2 |
| RA-DIT (2023) | 46.2 | 87.4 | 20.3 | 34.9 | **41.7** | 14.8 |
| RAG-DDR (w/ 1-Round) | 50.7 | 88.2 | 25.1 | 37.3 | 37.3 | 14.9 |
| RAG-DDR (w/ 2-Round) | **52.1** | **89.6** | **27.3** | **39.0** | 40.6 | **16.8** |

which optimizes the RAG system using the instruct-tuning method. In our experiments, we reimplement REPLUG and RA-DIT baselines and don't finetune the retriever during our reproduction process, as the retriever we use has already been trained with massive supervised data and is sufficiently strong.

**Implementation Details**. In our experiments, we employ Minicpm-2.4B-sft (Hu et al., 2024) and Llama3-8B-Instruct (Touvron et al., 2023) as backbone models to construct the generation modules and employ Llama3-8B-Instruct (Touvron et al., 2023) to build the knowledge refinement module. More training implementation details of RAG-DDR are presented in Appendix A.1.

## 5 EVALUATION RESULTS

In this section, we first evaluate the performance of different RAG methods and then conduct ablation studies to show the effectiveness of different training strategies. Then, we examine the effectiveness of DDR training strategies on the generation module ($V_{\text{Gen}}$) and explore how it balances internal and external knowledge through DDR. Finally, we present several case studies.

### 5.1 OVERALL PERFORMANCE

The performance of various RAG models is presented in Table 1. As shown in the evaluation results, RAG-DDR significantly outperforms these baseline models on all datasets. It achieves improvements of 7% compared to the Vanilla RAG model when using MiniCPM-2.4B and Llama3-8B to construct the generation module ($V_{\text{Gen}}$).

Compared with LLM w/o RAG, Vanilla RAG and REPLUG significantly enhance LLM performance on most knowledge-intensive tasks, indicating that external knowledge effectively improves the accuracy of generated responses. However, the performance of RAG models decreases on dialogue tasks, showing that LLMs can also be misled by the retrieved documents. Unlike these zero-shot methods, RA-DIT provides a more effective approach for guiding LLMs to filter out noise from retrieved content and identify accurate clues for answering questions. Nevertheless, RA-DIT still underperforms compared to Vanilla RAG on certain knowledge-intensive tasks, such as NQ and HotpotQA, showing that overfitting to golden answers is less effective for teaching LLMs to capture essential information for generating accurate responses. In contrast, RAG-DDR surpasses RA-DIT on almost all tasks, particularly with smaller LLMs (MiniCPM-2.4B), achieving a 5% improvement. This highlights the generalization capability of our DDR training method, enabling LLMs of varying scales to utilize external knowledge through in-context learning effectively.

Table 2: Ablation Study. Both Vanilla RAG (w/o $V_{KR}$) and Vanilla RAG are evaluated in a zero-shot setting without any fine-tuning. We then use DDR to optimize the knowledge refinement module ($V_{KR}$), the generation module ($V_{Gen}$), and both modules, resulting in three models: RAG-DDR (Only Training $V_{KR}$), RAG-DDR (Only Training $V_{Gen}$) and RAG-DDR (All Training).

| Method | Open-Domain QA | | | Multi-Hop QA | Slot Filling | Dialogue |
|---|---|---|---|---|---|---|
| | NQ | TriviaQA | MARCO QA | HotpotQA | T-REx | WoW |
| *MiniCPM-2.4B* | | | | | | |
| Vanilla RAG (w/o $V_{KR}$) | 42.1 | 78.0 | 16.6 | 24.9 | 22.0 | 15.1 |
| Vanilla RAG | 42.2 | 79.5 | 16.7 | 26.7 | 22.1 | 14.4 |
| RAG-DDR (Only Training $V_{KR}$) | 42.5 | 79.6 | 16.8 | 27.3 | 21.9 | 15.8 |
| RAG-DDR (Only Training $V_{Gen}$) | 46.8 | 81.7 | **28.3** | 31.2 | 32.2 | 17.0 |
| RAG-DDR (All Training) | **47.0** | **82.7** | 28.1 | **32.5** | **32.6** | **17.2** |
| *Llama3-8B* | | | | | | |
| Vanilla RAG (w/o $V_{KR}$) | 45.4 | 83.2 | 20.8 | 28.5 | 26.6 | 12.3 |
| Vanilla RAG | 46.2 | 84.0 | 20.6 | 30.1 | 26.9 | 12.5 |
| RAG-DDR (Only Training $V_{KR}$) | 46.8 | 84.7 | 20.7 | 30.7 | 28.5 | 12.5 |
| RAG-DDR (Only Training $V_{Gen}$) | 50.2 | 87.8 | **25.2** | 36.9 | 36.2 | 14.8 |
| RAG-DDR (All Training) | **50.7** | **88.2** | 25.1 | **37.3** | **37.3** | **14.9** |

## 5.2 ABLATION STUDIES

As shown in Table 2, we conduct ablation studies to explore the role of different RAG modules and evaluate different training strategies using DDR.

This experiment compares five models, utilizing MiniCPM-2.4B and Llama3-8B to construct the generation module. The Vanilla RAG (w/o $V_{KR}$) relies solely on the generation module ($V_{Gen}$) to produce answers based on the query and retrieved documents. The Vanilla RAG adds a knowledge refinement module ($V_{KR}$) to filter the retrieved documents and then feeds query and filtered documents to the generation module ($V_{Gen}$). RAG-DDR (Only Training $V_{KR}$) indicates that we tune the Vanilla RAG model using DDR by only optimizing the knowledge refinement module ($V_{KR}$). RAG-DDR (Only Training $V_{Gen}$) only optimizes the generation module ($V_{Gen}$). RAG-DDR (All Training) optimizes both $V_{KR}$ and $V_{Gen}$ modules.

Compared with the Vanilla RAG (w/o $V_{KR}$), Vanilla RAG improves the performance on almost all evaluation tasks, demonstrating the effectiveness of the knowledge refinement module in improving the accuracy of LLM responses. In contrast, RAG-DDR (Only Training $V_{Gen}$) shows greater improvements over than Vanilla RAG, indicating that the primary effectiveness of RAG-DDR comes from optimizing the generation module ($V_{Gen}$) through DDR. When we begin with the RAG-DDR (Only Training $V_{Gen}$) model and subsequently optimize the knowledge refinement module, the performance is further improved. It shows that filtering noise from retrieved documents using feedback from the generation module is effective, which is also observed in previous work (Yu et al., 2023b; Izacard & Grave, 2020). However, the improvements from optimizing the knowledge refinement modules are limited, highlighting that enhancing the generation module's ability to leverage external knowledge is more critical for the existing RAG system.

## 5.3 CHARACTERISTICS OF THE GENERATION MODULE IN RAG-DDR

In this experiment, we explore the characteristics of the generation module ($V_{Gen}$) by employing various training strategies, including zero-shot (Vanilla RAG), the SFT method (RA-DIT), and DDR (RAG-DDR). As illustrated in Figure 2, we present the performance of $V_{Gen}$ w/o RAG and $V_{Gen}$ w/ RAG. These experiments evaluate $V_{Gen}$'s ability to memorize knowledge and utilize external knowledge. Additionally, we report the average length of responses generated by $V_{Gen}$ module.

As shown in Figure 2(a), we compare the performance of the generation module that relies solely on internal knowledge of parametric memory. Compared to the Vanilla RAG model, RA-DIT demonstrates a decline in performance on the NQ and HotpotQA tasks. Such a phenomenon reveals that the model loses previously acquired knowledge while learning new information during SFT (Luo et al., 2023b). In contrast, DDR not only outperforms the RA-DIT method but also achieves consistent improvements over the Vanilla RAG model across all evaluation tasks. This indicates that DDR can help LLMs learn more factual knowledge during training while also preventing the loss of previously memorized information through a reinforcement learning-based training approach. Then

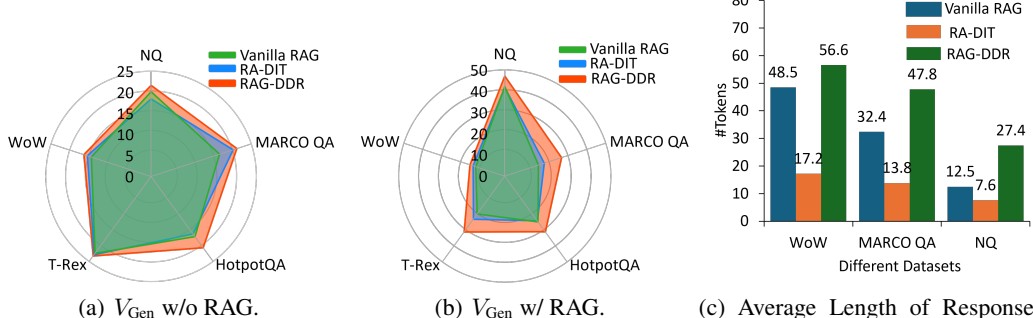

(a) $V_{\text{Gen}}$ w/o RAG.  (b) $V_{\text{Gen}}$ w/ RAG.  (c) Average Length of Responses Generated by $V_{\text{Gen}}$.

Figure 2: Characteristics of the Generation Module in RAG Optimized with Different Training Strategies. We use MiniCPM-2.4B to build the generation module ($V_{\text{Gen}}$) and then train it using different strategies. The performance of $V_{\text{Gen}}$ is shown in a zero-shot setting, along with the generation module optimized using the RA-DIT and DDR methods.

Table 3: Experimental Results on Evaluating the Knowledge Usage Ability of the Generation Module ($V_{\text{Gen}}$) of Different RAG Models.

| Method | Has-Answer | | | Miss-Answer | | | Internal Knowledge | | |
|---|---|---|---|---|---|---|---|---|---|
| | NQ | HotpotQA | T-REx | NQ | HotpotQA | T-REx | NQ | HotpotQA | T-REx |
| *MiniCPM-2.4B* | | | | | | | | | |
| LLM w/o RAG | 27.6 | 26.7 | 36.6 | **2.6** | 11.7 | 4.1 | 100.0 | 100.0 | 100.0 |
| Vanilla RAG | 59.1 | 51.7 | 36.8 | 1.1 | 9.1 | 2.5 | 71.1 | 70.1 | 60.9 |
| RA-DIT | 58.3 | 47.6 | 41.7 | 1.7 | 10.8 | 4.2 | 76.9 | 73.7 | 73.4 |
| RAG-DDR (w/ 1-Round) | **65.5** | 56.9 | **52.6** | 2.4 | 10.8 | **5.9** | 82.9 | 81.0 | **78.5** |
| RAG-DDR (w/ 2-Round) | 65.3 | **59.4** | 51.7 | 2.4 | **14.2** | 5.5 | **84.0** | **82.3** | 74.8 |
| *Llama3-8B* | | | | | | | | | |
| LLM w/o RAG | 46.4 | 44.7 | 25.5 | **6.7** | 15.9 | 3.7 | 100.0 | 100.0 | 100.0 |
| Vanilla RAG | 64.2 | 58.0 | 45.5 | 2.9 | 10.5 | 3.0 | 80.0 | 67.3 | 66.3 |
| RA-DIT | 64.1 | 59.7 | 65.3 | 3.4 | 17.9 | **10.8** | 81.0 | 79.2 | 77.9 |
| RAG-DDR (w/ 1-Round) | 69.5 | 64.3 | 59.9 | 4.1 | 18.0 | 5.4 | 88.4 | 82.0 | 79.2 |
| RAG-DDR (w/ 2-Round) | **71.6** | **66.2** | **65.4** | 5.6 | **18.9** | 7.7 | **89.3** | **83.9** | **89.7** |

we feed retrieved documents to the generation module and show the generation performance in Figure 2(b). The evaluation results indicate that RA-DIT marginally outperforms the Vanilla RAG model, while RAG-DDR significantly improves generation accuracy by utilizing factual knowledge from the retrieved documents. Additional experiments showing the general capabilities of RAG-DDR are presented in Appendix A.7.

Finally, we show the average length of responses generated by $V_{\text{Gen}}$ in Figure 2(c). Compared to the Vanilla RAG model, the average length of responses generated by RA-DIT decreases significantly, indicating that the SFT training method tends to cause LLMs to overfit the training dataset. On the contrary, RAG-DDR shows a more similar length distribution with the Vanilla RAG model, enabling the model to generate responses of a more appropriate length. It demonstrates that training LLMs to learn data preferences from generated responses can help align the output format of RAG models more closely with that of the original LLMs.

### 5.4 EFFECTIVENESS OF RAG-DDR IN USING EXTERNAL KNOWLEDGE

In this section, we investigate the capability of the generation module $V_{\text{Gen}}$ in the RAG model to leverage external knowledge for response generation. We first evaluate the ability of $V_{\text{Gen}}$ to balance the internal and external knowledge. Next, we evaluate the denoising ability by $V_{\text{Gen}}$ feeding additional unrelated documents as the external knowledge.

As shown in Table 3, we first show the effectiveness of $V_{\text{Gen}}$ in balancing internal and external knowledge during producing responses. We compare three training strategies: zero-shot (Vanilla RAG), SFT (RA-DIT), and DDR (RAG-DDR). Our experiment establishes three testing scenarios to evaluate the effectiveness of different RAG models by categorizing the evaluation data into three distinct

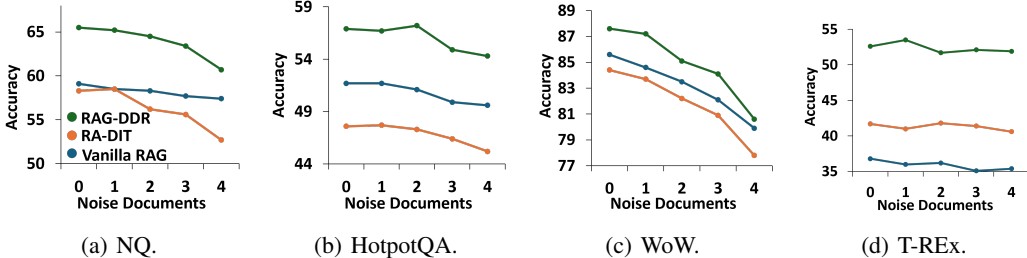

(a) NQ.  (b) HotpotQA.  (c) WoW.  (d) T-REx.

Figure 3: Effectiveness of Different RAG Models in Defending Noisy Information. We use MiniCPM-2.4B to build the generation module ($V_{\text{Gen}}$). Then we retain one informative document and randomly replace $n$ top-retrieved documents with noisy ones.

scenarios: Has-Answer, Miss-Answer, and Internal Knowledge. The Has-Answer scenario indicates that the retrieved documents contain the golden answer, which can help the generation module to answer the question accurately. The Miss-Answer scenario indicates that the retrieved documents do not contain the golden answer and fail to provide sufficient support for LLMs to generate accurate responses. Lastly, the Internal Knowledge scenario further evaluates the ability of LLMs in dealing with the conflict between internal and external knowledge. The test cases of the Internal Knowledge scenario indicate that LLMs can generate accurate answers using only parametric memory, whereas RAG models may produce incorrect responses.

In the Has-Answer scenario, RAG-DDR and RA-DIT outperform the Vanilla RAG model on all datasets. This indicates that training the generation module enables it to capture essential knowledge facts, improving the accuracy of the generated responses. Compared with RA-DIT, RAG-DDR achieves consistent improvements over the Vanilla RAG model, which shows that DDR can better generalize the external knowledge usage ability to different tasks. In the Miss-Answer scenario, all RAG models perform significantly worse compared to the Has-Answer scenario, showing that the retrieved knowledge fails to provide sufficient information for generating accurate responses. Compared to the Vanilla RAG, RAG-DDR and RA-DIT mitigate the performance drop caused by incorporating retrieved documents by fine-tuning the generation module, illustrating their ability to effectively leverage internal knowledge and avoid being misled by irrelevant information. For the Internal Knowledge scenario, we evaluate the ability of RAG models in handling the conflict from external knowledge. The Vanilla RAG model decreases the generation accuracy more than 20% on all tasks, showing that the knowledge conflict can significantly affect the generation result. DDR exhibits strong effectiveness in mitigating the knowledge conflict in the Vanilla RAG model, resulting in a reduction of more than 10% in performance drop. This indicates that DDR effectively balances the internal and external knowledge, facilitating the robustness of the RAG model.

In the second experiment, we extend the Has-Answer setting and further investigate the denoising effectiveness of $V_{\text{Gen}}$ by adding the different number of noise documents. As shown in Figure 3, we increase the noise by randomly replacing $n$ documents from the top-5 retrieved set with the last $n$ documents from the top-100 retrieved candidates, while ensuring that the ground truth document remains in the set. RA-DIT exhibits inconsistent performance and degrades in some RAG scenarios. In contrast, RAG-DDR consistently outperforms the Vanilla RAG model, maintaining a consistent improvement even as the number of noisy documents increases. It further confirms the effectiveness of our RAG-DDR approach in defending noise.

### 5.5 CASE STUDIES

In Table 4, we present three randomly selected examples from the NQ, HotpotQA, and TriviaQA datasets to show the generated responses and evaluate the effectiveness of the RAG-DDR model.

For the first case, the query asks about the "daylight savings time of Canada" and the retrieved documents contain detailed information about when the daylight savings time begins each year in Canada. However, since the exact date of daylight savings time changes annually, the most accurate answer is "the second Sunday in March". RAG-DDR shows its effectiveness in accurately answer-

Table 4: Case Studies. These matching phrases are indicated in **Red** if the model answers correctly and in **Blue** if it answers incorrectly. We present the generated responses of different RAG models, which use MiniCPM-2.4B to build the generation module ($V_{\text{Gen}}$).

---

**Case 1:** When does Canada switch to daylight savings time
**Document:** Daylight Saving Time Canada. Daylight Saving Time in Canada. In most of Canada Daylight Saving Time begins at 2:00 a.m. local time on **the second Sunday in March**. ... Canada Daylight Saving Time Start and End Dates 2020 March 8 November 1 2019 **March 10** November ... November 6 2010 **March 14** November 7 2009 March 08 November 1 ...

**Vanilla LLMs:** Canada does not switch to or from daylight savings time. Instead, the clocks are automatically set forward by one hour during the spring.
**Vanilla RAG:** **March 14**    **RA-DIT:** **March 8**
**RAG-DDR:** Canada switches to Daylight Saving Time on **the second Sunday of March** and returns to Standard Time on the first Sunday of November.

---

**Case 2:** The manager who recruited David Beckham managed Manchester United during what timeframe
**Document:** Ferguson's unprecedented achievements at Aberdeen led to managerial offers from some of the most prestigious clubs in Europe over the years, and he signed with Manchester in November **1986**. ... **Ferguson retired at the end of the 2012–13 Premier League season** but stayed on with Man U in a...

**Vanilla LLMs:** 1992–2003    **Vanilla RAG:** **1986**–1993    **RA-DIT:** 1992 to **2013**
**RAG-DDR:** The football manager who recruited David Beckham was Sir Alex Ferguson, and he managed Manchester United from **1986 to 2013**.

---

**Case 3:** Following success at the 1979 election whose party conference speeech included the lines 'you turn if you want to, the lady's not for turning'?
**Document:** The lady's not for turning - Wikipedia The lady's not for turning From Wikipedia, the free encyclopedia Jump to navigation Jump to search 1980 **Margaret Thatcher** speech ... a phrase used by Margaret Thatcher, then Prime Minister, in her speech to **the Conservative Party Conference**...

**Vanilla LLMs:** The speaker in question is **Margaret Thatcher**, who was the leader of the Conservative Party and later became the Prime Minister of the United Kingdom.
**Vanilla RAG:** **Conservative Party Conference**    **RA-DIT:** **The Conservative Party**
**RAG-DDR:** The 1979 Conservative Party conference speech by **Margaret Thatcher** included the lines "you turn if you want to, the lady's not for turning".

---

ing the question, while both Vanilla RAG and RA-DIT are misled by noisy information such as "14th March" and "10th March", leading to incorrect responses. This demonstrates that RAG-DDR model has the ability to distinguish the most accurate knowledge from ambiguous or misleading information in the retrieved documents. In the second case, the model must integrate multiple pieces of knowledge from the provided documents to answer the question. While the Vanilla RAG model and RAG-DIT only correctly answer half of the questions using partial knowledge, RAG-DDR successfully identifies the correct start time and end time. This indicates that RAG-DDR has a stronger capacity to integrate factual knowledge from different document segments. As shown in the third case, Vanilla LLM can answer the question correctly only depending on the parametric memory. Nevertheless, both Vanilla RAG and RA-DIT are misled by the confusing information from these retrieved documents and generate the response "the Conservative Party Conference", which is entirely unrelated to the given question. In contrast, RAG-DDR accurately follows the intent of the question for generating the response, demonstrating the ability of RAG-DDR to mitigate the negative influence of external knowledge.

## 6    CONCLUSION

This paper proposes Differentiable Data Rewards (DDR), aiming at end-to-end optimizing the Retrieval-Augmented Generation (RAG) model using the DPO method. DDR optimizes each agent by collecting the reward in a rollout way and aligns data preferences among these communicative agents. We build a two-agent RAG system and optimize it using DDR to implement the RAG-DDR model. Our experiments demonstrate that DDR helps the generation module produce responses of an appropriate length and avoids overfitting the training signals during SFT. Our further analyses reveal that the DDR optimized generation model can better capture key information from retrieved documents and mitigate the conflict between external knowledge and parametric memory.

ACKNOWLEDGMENTS

This work is partly supported by the Natural Science Foundation of China under Grant (No. 62206042, No. 62137001, and No. 62272093), CCF-zhipu Large Model Innovation Fund (No. 202403).

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

Table 5: Data Statistics.

| Split | Task | Dataset | Metric | Total |
|---|---|---|---|---|
| Training | Open-Domain QA | Commonsense QA (2019) | Accuracy | 4,200 |
| | | Math QA (2019) | Accuracy | 4,200 |
| | | Web Questions (2013) | Accuracy | 3,778 |
| | | Wiki QA (2015) | Rouge-L | 1,040 |
| | | Yahoo! Answers QA | Rouge-L | 4,200 |
| | | MARCO QA (2016) | Rouge-L | 4,200 |
| | Reasoning | Algebra QA with Rationales (2017) | Accuracy | 2,727 |
| | | Explanations for CommonsenseQ (2021) | Accuracy | 4,200 |
| | | Grade School Math 8K (2021) | Accuracy | 4,200 |
| | | StrategyQA (2021) | Accuracy | 2,060 |
| Evaluation | Open-domain QA | Natural Questions (2019) | Accuracy | 2,837 |
| | | TriviaQA (2017) | Accuracy | 5,359 |
| | | MARCO QA (2016) | Rouge-L | 3,000 |
| | Multi-Hop QA | HotpotQA (2018) | Accuracy | 5,600 |
| | Slot Filling | T-REx (2018) | Accuracy | 5,000 |
| | Dialogue | Wizard of Wikipedia (2019) | F1 | 3,000 |

# A APPENDIX

## A.1 ADDITIONAL EXPERIMENTAL DETAILS

In this subsection, we first outline the process of constructing the training data. Then we show the prompt templates used in our experiments. Then, we describe the maximum generation length of RAG-DDR during inference. Finally, we provide details on the implementation of RAG-DDR training. To ensure a fair comparison, we train both RA-DIT and RAG-DDR using the same dataset, hyperparameters, and number of epochs. For Self-RAG, we adhere to it respective training data formats and hyperparameter settings.

**Data Preprocessing for DDR.** The quantity of our training and evaluation data, along with the corresponding evaluation metrics, are presented in Table 5. Then we describe the details of data preprocessing during training both knowledge refinement ($V_{KR}$) and generation ($V_{Gen}$) modules using our DDR method.

Optimizing the knowledge refinement module is a rollout process, which individually feeds the top-100 retrieved documents into the generation module ($V_{Gen}$) with the query to calculate the reward. Then, we apply the evaluation metrics shown in Table 5 to calculate the reward scores, identifying the documents that result in the highest scores as positive documents and those with the lowest scores as negative ones. Finally, we obtain the triplet data {query, positive/negative documents, "YES"/"NO"}. To construct the dataset for training the generation module, we concatenate the refined documents with the query and then feed them into the generation module for sampling responses. We apply five different temperature settings $(0.5, 0.6, 0.7, 0.8, 0.9)$ to sample responses and conduct the five-round sampling for each temperature. Afterward, we compute the reward score for each output using evaluation metrics to identify the positive and negative responses for each query. This process yields the triplet data {query, positive response, negative response}.

**Maximum Generation Length.** For the evaluation dataset, we set the maximum generation length for MARCO QA task to 100, while the maximum generation length for the rest tasks is set to 32.

**Prompt Templates.** For RA-DIT, we use the same instruction tuning template as Lin et al. (2023) and leverage top-5 retrieved documents as augmented knowledge during training. For Self-RAG, we use the same instruction tuning template as Asai et al. (2023) during training. Then we describe the prompt templates used in RAG-DDR. As shown in Figure 4, we refer to the prompt designs of RA-DIT (Lin et al., 2023) and Self-RAG (Asai et al., 2023) to conduct tailored task prompts for different LLMs, helping LLMs generate better responses. In addition, we design separate prompts for LLMs w/ RAG and LLMs w/o RAG:{Background:{*Documents*}\n {*Instruction*}} and {*Instruction*}, where *Documents* indicates the external documents provided for LLMs and *Instruction* represents the task instructions. As illustrated in Figure 5, we design the prompts for

```
┌─ Prompts of Different Tasks (MiniCPM-2.4B) ─┐
│ [TASK]: Commonsense QA, Math QA             │
│ [Instruction]: The following is multiple    │
│ choice question. Please choose the best     │
│ answer choice which can answer the following│
│ question.                                   │
│ {Question}                                  │
│ Answer:                                     │
│                                             │
│ [TASK]: Web Questions, Wiki QA, Yahoo! QA,  │
│ MS MARCO QA, NQ, TriviaQA, HotpotQA         │
│ [Instruction]: Q: {Question}                │
│ A:                                          │
│                                             │
│ [TASK]: Algebra QA with Rationales,         │
│ Explanations for CommonsenseQ               │
│ [Instruction]: Please answer multiple choice│
│ question and choose the best answer choice  │
│ first. Then give your explanation between   │
│ [<COT] and [COT>].                          │
│ question: {Question}                        │
│ Answer:                                     │
│                                             │
│ [TASK]: Grade School Math 8K, Strategy QA   │
│ [Instruction]: Please answer the question.  │
│ Then give your explanation between [<COT]   │
│ and [COT>].                                 │
│ question: {Question}                        │
│ Answer:                                     │
│                                             │
│ [TASK]: Soft Filling                        │
│ [Instruction]: Given the input format       │
│ 'Subject Entity [SEP] Relationship Type,'   │
│ predict the target entity.                  │
│ {Question}                                  │
│ Answer:                                     │
│                                             │
│ [TASK]: Dialogue                            │
│ [Instruction]: Q: {Question_1}              │
│ A: {Answer_1}                               │
│ . . .                                       │
│ Q: {Question_n}                             │
│ A:                                          │
└─────────────────────────────────────────────┘
```

```
┌─── Prompts of Different Tasks (Llama3-8B) ───┐
│ [TASK]: Commonsense QA, Math QA              │
│ [Instruction]: Please answer the multiple    │
│ choice questions below and output only the   │
│ choice.                                      │
│ {Question}                                   │
│ Answer:                                      │
│                                              │
│ [TASK]: Web Questions, Wiki QA, Yahoo! QA,   │
│ MS MARCO QA , NQ, TriviaQA, HotpotQA         │
│ [Instruction]: Q: {Question}                 │
│ A:                                           │
│                                              │
│ [TASK]: Algebra QA with Rationales,          │
│ Explanations for CommonsenseQ                │
│ [Instruction]: Please answer the multiple    │
│ choice questions below and output only the   │
│ choice.                                      │
│ {Question}                                   │
│ Answer:                                      │
│                                              │
│                                              │
│ [TASK]: Grade School Math 8K, Strategy QA    │
│ [Instruction]: Please answer the question and│
│ only output the answer. Then give your       │
│ explanation between [<COT] and [COT>].       │
│ question: {Question}                         │
│ Answer:                                      │
│                                              │
│ [TASK]: Soft Filling                         │
│ [Instruction]: Given the input format        │
│ 'Subject Entity [SEP] Relationship Type,'    │
│ predict the target entity.                   │
│ {Question}                                   │
│ Answer:                                      │
│                                              │
│ [TASK]: Dialogue                             │
│ [Instruction]: Q: {Question_1}               │
│ A: {Answer_1}                                │
│ . . .                                        │
│ Q: {Question_n}                              │
│ A:                                           │
└──────────────────────────────────────────────┘
```

(a) MiniCPM-2.4B.          (b) Llama3-8B.

Figure 4: Prompt Templates of Different Training and Evaluating Tasks.

```
┌────────── Prompts of Knowledge Refinement ──────────┐
│                                                     │
│ [Instruction]: Given the following question and     │
│ context, return YES if the context is relevant to   │
│ the question and NO if it isn't.                     │
│                                                     │
│ > Question: {question}                              │
│ > Context:                                          │
│ >>>                                                 │
│ {context}                                           │
│ >>>                                                 │
│ > Relevant (YES / NO):                              │
│                                                     │
└─────────────────────────────────────────────────────┘
```

Figure 5: Prompt Templates Used for Knowledge Refinement.

the knowledge refinement tasks by referring to LangChain[1], enabling LLMs to correctly generate "YES" or "NO" to retain or discard retrieved documents.

**Training Details.** For DDR training, we use automatic metrics such as Rouge-L and Accuracy to calculate the reward and set $\beta = 0.1$. The learning rate is set to 5e-5, and each model is trained for one epoch. For the generation module, we feed 5 retrieved passages as external knowledge for augmenting the generation process. To optimize both the knowledge refinement module and generation module, we use LoRA (Hu et al., 2022) for efficient training.

---

[1] https://github.com/langchain-ai/langchain

Table 6: Overall Performance of RAG-DDR and Additional Baselines. The evaluation scores of InstructRAG on the MARCO QA and WoW tasks are usually low. The reason may lie in that the responses generated by InstructRAG are long and contain complex reasoning processes, making it difficult to conduct a fair evaluation for these string matching based evaluation metrics.

| Method | Open-Domain QA | | | Multi-Hop QA | Slot Filling | Dialogue |
|---|---|---|---|---|---|---|
| | NQ | TriviaQA | MARCO QA | HotpotQA | T-REx | WoW |
| *Short Generation Length* | | | | | | |
| Filco (2023) | 46.6 | 81.9 | 14.6 | 25.3 | **44.1** | 14.4 |
| InstructRAG (2024) | 4.2 | 21.1 | 14.3 | 11.0 | 17.1 | 9.2 |
| RAG-DDR (w/ 2-Round) | **52.1** | **89.6** | **27.3** | **39.0** | 40.6 | **16.8** |
| *Long Generation Length* | | | | | | |
| InstructRAG (2024) | **60.8** | **92.0** | 12.4 | 42.3 | 43.1 | 5.3 |
| RAG-DDR (w/ 2-Round) | 58.0 | 91.9 | **26.1** | **43.9** | **45.6** | **10.9** |

Table 7: Additional Ablation Study Results on RAG-DDR.

| Method | Open-Domain QA | | | Multi-Hop QA | Slot Filling | Dialogue |
|---|---|---|---|---|---|---|
| | NQ | TriviaQA | MARCO QA | HotpotQA | T-REx | WoW |
| *MiniCPM-2.4B* | | | | | | |
| Independent Tuning | 46.6 | 82.0 | 28.0 | 32.3 | 31.7 | 17.3 |
| RAG-DDR ($V_{KR}$ First) | **47.4** | **83.2** | 27.2 | **33.3** | **35.0** | **17.3** |
| RAG-DDR ($V_{Gen}$ First) | 47.0 | 82.7 | **28.1** | 32.5 | 32.6 | 17.2 |
| *Llama3-8B* | | | | | | |
| Independent Tuning | 49.9 | 88.3 | 25.1 | 37.4 | 37.0 | 14.9 |
| RAG-DDR ($V_{KR}$ First) | 50.4 | **88.6** | **25.6** | 36.9 | 36.7 | 14.9 |
| RAG-DDR ($V_{Gen}$ First) | **50.7** | 88.2 | 25.1 | **37.3** | **37.3** | **14.9** |

## A.2 ADDITIONAL BASELINE COMPARISON RESULTS

This section presents the comparison results between RAG-DDR (w/ 2-Round) and other baseline models. We employ InstructRAG (Wei et al., 2024) and Filco (Wang et al., 2023) as baseline models. InstructRAG guides the LLM to self-synthesize denoising instruction tuning data and trains the generation module to denoise the retrieved documents. Filco finetunes Flan-T5-XL (Chung et al., 2024) as a context filtering module to refine the retrieved documents. Specifically, the responses generated by InstructRAG include long reasoning and analysis, thus InstructRAG sets the maximum generation length as 4,096. For a fair comparison, we compare the performance of RAG-DDR and InstructRAG on the two maximum generation length settings: the short generation length (MARCO QA task is 100 and the rest tasks are 32), and the Long generation length (4,096). In our experiments, we employ Llama3-8B-Instruct as the knowledge refinement and generation modules.

As shown in Table 6, RAG-DDR outperforms Filco in almost all tasks. This suggests that the SFT method leads modules in the RAG system to overfit the training signals, making it less effective to optimize the entire RAG system to generate accurate responses. In contrast, in the long generation length setting, InstructRAG slightly outperforms RAG-DDR on NQ and TriviaQA tasks but performs slightly worse than RAG-DDR on T-REX and HotpotQA tasks. This illustrates that high-quality synthetic fine-tuning data can effectively optimize the performance of the generation module. However, in the short generation length setting, InstructRAG performs significantly worse than RAG-DDR, which indicates that compared to InstructRAG, RAG-DDR can generate accurate answers with shorter responses, without complex and long reasoning processes.

## A.3 ADDITIONAL ABLATION STUDIES ON RAG-DDR

As shown in Table 7, we conduct additional ablation studies to explore the effectiveness of different training strategies: Independent Tuning, RAG-DDR ($V_{KR}$ First) and RAG-DDR ($V_{Gen}$ First). Independent Tuning indicates that the knowledge refinement module $V_{KR}$ and generation module $V_{Gen}$ are trained independently. RAG-DDR ($V_{KR}$ First) and RAG-DDR ($V_{Gen}$ First) are cascaded optimization models, indicating that we first train $V_{KR}$ or $V_{Gen}$ and subsequently optimize the other module by initializing the RAG model with the already optimized module.

Table 8: Effectiveness of RAG-DDR with More Agents. Llama3-8B is used as the backbone model for both the knowledge refinement and summarization modules, while MiniCPM-2.4B is used as the backbone model for the generation module.

| Method | Open-Domain QA | | | Multi-Hop QA | Slot Filling | Dialogue |
|---|---|---|---|---|---|---|
| | NQ | TriviaQA | MARCO QA | HotpotQA | T-REx | WoW |
| Vanilla RAG (w/o $V_{Sum}$) | 42.2 | 79.5 | 16.7 | 26.7 | 22.1 | 14.4 |
| Vanilla RAG | 44.1 | 82.3 | 17.5 | 28.9 | 25.4 | 15.4 |
| RAG-DDR (Only Training $V_{Gen}$) | 46.6 | 83.6 | **26.9** | 32.1 | 28.3 | 16.9 |
| RAG-DDR (Training $V_{Gen}$ & $V_{Sum}$) | 47.6 | 83.9 | 26.4 | 32.0 | 28.6 | **17.0** |
| RAG-DDR (All Training) | **47.9** | **84.8** | 26.4 | **32.9** | **28.7** | 16.9 |

Table 9: Effectiveness of RAG-DDR Using Large-Scale LLMs. In our experiments, we utilize Qwen2.5-14B-Instruct as the generation module and Llama3-8B-Instruct as the refinement module.

| Method | Open-Domain QA | | | Multi-Hop QA | Slot Filling | Dialogue |
|---|---|---|---|---|---|---|
| | NQ | TriviaQA | MARCO QA | HotpotQA | T-REx | WoW |
| Vanilla RAG | 48.1 | 81.1 | 18.5 | 29.4 | 37.3 | 14.3 |
| RAG-DDR | **51.2** | **84.1** | **23.0** | **35.1** | **44.6** | **16.6** |

Compared to Independent Tuning, the effectiveness of RAG-DDR is enhanced on all tasks. It shows that Independent Tuning results in misalignment of data preferences between the optimized modules, thereby impacting the overall performance of the RAG system. In contrast, the performance of RAG-DDR ($V_{KR}$ First) and RAG-DDR ($V_{Gen}$ First) is indistinguishable across different datasets and $V_{Gen}$, demonstrating that the DDR method is robust to the training orders.

## A.4 EFFECTIVENESS OF RAG-DDR WITH MORE AGENTS

In this experiment, we extend RAG-DDR to more agents to evaluate its effectiveness in optimizing complex RAG systems. Specifically, we utilize the knowledge refinement ($V_{KR}$), summarization ($V_{Sum}$), and generation ($V_{Gen}$) modules to construct a new RAG system. In this system, the knowledge refinement module refines the retrieved documents $D$ to get the refined documents $\tilde{D}$, the summarization module summarizes the refined documents $\tilde{D}$ into a concise summary $\tilde{S}$ based on the query $q$, and the generation module uses this summary $\tilde{S}$ to answer the query $q$. This system can be represented as a three-agent system:

$$\{q, D\} \rightsquigarrow V_{KR} \xrightarrow{\{q,\tilde{D}\}} V_{Sum} \xrightarrow{\{q,\tilde{S}\}} V_{Gen} \rightsquigarrow y_{Gen}, \tag{9}$$

where $\tilde{D} \subseteq D$. $\{q, D\} \rightsquigarrow$ and $\rightsquigarrow y_T$ represent sending the input $\{q, D\}$ to the RAG system and getting the final output $y_T$. In our implementation, we use Llama3-8B-Instruct as the backbone model to construct the knowledge refinement and summarization modules, and Minicpm-2.4B-sft as the backbone model to construct the generation module.

As shown in Table 8, we employ five models: Vanilla RAG (w/o $V_{Sum}$, Vanilla RAG, RAG-DDR (Only Training $V_{Gen}$), RAG-DDR (Training $V_{Gen}$ & $V_{Sum}$) and RAG-DDR (All Training) to evaluate the effectiveness of the DDR method in RAG systems with more agents. RAG-DDR (Only Training $V_{Gen}$) indicates that we tune the Vanilla RAG using DDR by only optimizing the generation module ($V_{Gen}$). RAG-DDR (Training $V_{Gen}$ & $V_{Sum}$) indicates that we use DDR to optimize both the generation module ($V_{Gen}$) and summarization module ($V_{Sum}$) in Vanilla RAG. RAG-DDR (All Training) indicates that we optimize all three modules in the Vanilla RAG.

Compared to Vanilla RAG (w/o $V_{Sum}$), Vanilla RAG demonstrates performance improvements across all evaluation tasks, showing the effectiveness of the summarization module in extracting relevant information from multiple documents. In contrast, RAG-DDR (Only Training $V_{Gen}$) shows greater improvements than Vanilla RAG, indicating that the primary improvements of RAG-DDR come from optimizing the generation module ($V_{Gen}$). When we subsequently optimize the summarization and knowledge refinement module based on RAG-DDR (Only Training $V_{Gen}$), the performance is further improved. This indicates that DDR not only improves the performance of the RAG system but also exhibits strong scalability, allowing it to be extended to different RAG systems.

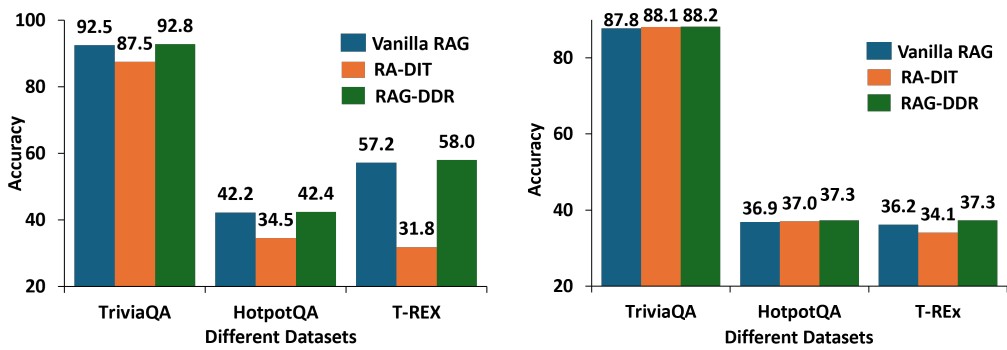

(a) The Accuracy of Knowledge Refinement.

(b) The Accuracy of Generation Using Different Refined Document Sets.

Figure 6: Effectiveness of the Knowledge Refinement Module ($V_{KR}$) Optimized Using Different Methods, Including Zero-shot (Vanilla RAG), SFT (RA-DIT), and DDR (RAG-DDR).

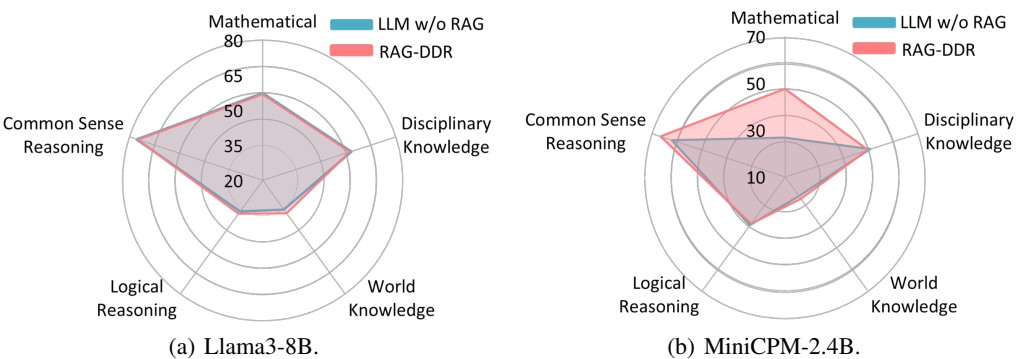

(a) Llama3-8B.

(b) MiniCPM-2.4B.

Figure 7: The Ability of the Generation Module ($V_{Gen}$) in RAG-DDR.

## A.5 EXTENDING RAG-DDR TO LARGER-SCALE LLMS

In this section, we explore the effectiveness of the DDR method in the RAG system that employs a larger-scale LLM as the backbone model of the generation module. Specifically, we utilize Qwen2.5-14B-Instruct (Yang et al., 2024) as the generation module and Llama3-8B-Instruct as the refinement module to build a new combination for the RAG system. As shown in Table 9, compared to vanilla RAG, RAG-DDR shows significant improvements across multiple tasks, highlighting the effectiveness of the DDR method in the RAG system with the larger-scale LLM.

## A.6 CHARACTERISTICS OF THE KNOWLEDGE REFINEMENT MODULE OF RAG-DDR

As shown in Figure 6, we explore the characteristics of the knowledge refinement module ($V_{KR}$) optimized using different methods, including zero-shot (Vanilla RAG), the SFT method (RA-DIT), and DDR (RAG-DDR).

For RA-DIT, we collect 30k pieces of data from the MARCO QA dataset to tune $V_{KR}$ module where each query has labeled positive and negative documents. We sample one positive document and one negative document for each query to construct the SFT dataset containing 60k pieces of data and fine-tune $V_{KR}$ module. The number of the SFT dataset is consistent with the amount of the RAG-DDR training dataset. The SFT dataset consists of triples in the form of {query, positive/negative documents, "YES"/"NO"}.

As shown in Figure 6(a), we calculate the accuracy of top-5 documents by Vanilla RAG, RA-DIT, and RAG-DDR. The retrieval accuracy evaluates whether or not the retained top-5 documents con-

Table 10: The Effectiveness of Generation Module Optimized with Different Training Strategies.

| Method | Open-Domain QA | | | Multi-Hop QA | Slot Filling | Dialogue |
|---|---|---|---|---|---|---|
| | NQ | TriviaQA | MARCO QA | HotpotQA | T-REx | WoW |
| *MiniCPM-2.4B* | | | | | | |
| Vanilla RAG | 42.2 | 79.5 | 16.7 | 26.7 | 22.1 | 14.4 |
| RA-DIT (Positive Label) | 40.9 | 77.2 | 19.9 | 22.0 | 25.3 | 14.7 |
| RA-DIT (Ground Truth) | 41.8 | 78.6 | 19.6 | 26.1 | 25.2 | 15.7 |
| RAG-DDR | **46.8** | **81.7** | **28.3** | **31.2** | **32.2** | **17.0** |
| *Llama3-8B* | | | | | | |
| Vanilla RAG | 46.2 | 84.0 | 20.6 | 30.1 | 26.9 | 12.5 |
| RA-DIT (Positive Label) | 45.8 | 86.4 | 23.0 | 31.4 | 25.3 | 13.2 |
| RA-DIT (Ground Truth) | 46.2 | 87.4 | 20.3 | 34.9 | **41.7** | 14.8 |
| RAG-DDR | **50.2** | **87.8** | **25.2** | **36.9** | 36.2 | **14.8** |

Table 11: The Average Length of Responses Generated by Generation Module Optimized with Different Training Strategies.

| Method | Open-Domain QA | | | Multi-Hop QA | Slot Filling | Dialogue |
|---|---|---|---|---|---|---|
| | NQ | TriviaQA | MARCO QA | HotpotQA | T-REx | WoW |
| *MiniCPM-2.4B* | | | | | | |
| Vanilla RAG | 12.5 | 4.7 | 32.4 | 5.6 | 4.4 | 48.5 |
| RA-DIT (Positive Label) | 19.2 | 8.3 | 34.6 | 13.7 | 6.7 | 32.8 |
| RA-DIT (Ground Truth) | 7.6 | 4.8 | 13.8 | 8.9 | 5.0 | 17.2 |
| RAG-DDR | 27.4 | 16.9 | 47.8 | 27.2 | 11.2 | 56.6 |
| *Llama3-8B* | | | | | | |
| Vanilla RAG | 49.6 | 25.0 | 85.4 | 43.9 | 34.4 | 98.5 |
| RA-DIT (Positive Label) | 42.1 | 14.8 | 72.2 | 24.0 | 3.9 | 72.2 |
| RA-DIT (Ground Truth) | 6.9 | 8.7 | 14.5 | 23.6 | 4.0 | 16.7 |
| RAG-DDR | 56.0 | 33.9 | 81.2 | 51.6 | 35.2 | 101.5 |

tain the ground truth. If $V_{\text{KR}}$ module discards all the documents retrieved, the accuracy is 0. RAG-DDR outperforms Vanilla RAG and RA-DIT on all tasks. It indicates that the DDR method can make $V_{\text{KR}}$ module accurately retain documents that contain the necessary knowledge to answer the query. As shown in Figure 6(b), we use the refined documents by different knowledge refinement module ($V_{\text{KR}}$) to augment the DDR trained generation module ($V_{\text{Gen}}$). DDR-RAG also outperforms other models, indicating that DDR can help better align data preferences between $V_{\text{Gen}}$ and $V_{\text{KR}}$ modules.

### A.7 THE GENERAL LLM ABILITY OF DDR OPTIMIZED GENERATION MODULE

In this experiment, we further explore the characteristics of the generation module ($V_{\text{Gen}}$) optimized using different methods, zero-shot (LLM w/o RAG) and DDR (RAG-DDR).

As shown in Figure 7, we compare the general ability of the generation module on several aspects: Mathematical (Liu et al., 2024a), Disciplinary Knowledge (Hendrycks et al., 2020), World Knowledge (Kwiatkowski et al., 2019), Logical Reasoning (Suzgun et al., 2022), and Common Sense Reasoning (Clark et al., 2018). These tasks are commonly used as benchmarks to assess the model's inherent capabilities (Touvron et al., 2023; Hu et al., 2024).

As shown in Figure 7(a), DDR enables Llama3-8B to maintain its strong language understanding and knowledge reasoning capabilities. The performance of MiniCPM-2.4B is shown in Figure 7(b). The evaluation results show that DDR significantly enhances the performance of the smaller parameter models, MiniCPM-2.4B, particularly in mathematical and common sense reasoning tasks. It illustrates that DDR not only preserves the original capabilities of LLMs but also offers some potential for enhancing their performance.

### A.8 THE IMPACT OF DIFFERENT TRAINING STRATEGIES ON THE GENERATION MODULE

In this experiment, we further investigate the impact of different training strategies on the generation module $V_{\text{Gen}}$'s ability to utilize external knowledge and the average length of responses generated by $V_{\text{Gen}}$ module. We compare four models in this experiment, including Vanilla RAG, RA-DIT

(Positive Label), RA-DIT (Ground Truth) and RAG-DDR. For RA-DIT (Positive Label), we employ the RA-DIT method to train the Vanilla RAG and regard the positive response in DDR training data as ground truth. For RA-DIT (Ground Truth), we employ the RA-DIT method to train the Vanilla RAG with the annotated labels.

As shown in Table 10, RAG-DDR consistently outperforms RA-DIT (Positive Label) and RA-DIT (Ground Truth) across different datasets, highlighting the effectiveness of the DDR method to enhance the ability of $V_{Gen}$ module to utilize external knowledge. In contrast, RA-DIT (Positive Label) performs worse than RA-DIT (Ground Truth), demonstrating that relying solely on positive samples for supervised fine-tuning is insufficient for effectively training LLMs.

Furthermore, we analyze the average length of the responses generated by $V_{Gen}$ module in Table 11. Compared to the RA-DIT (Ground Truth), RAG-DDR shows a more similar generation length distribution to the Vanilla RAG model, making the LLMs generate responses with a more appropriate length to answer the question. Notably, the response length of RA-DIT (Positive Label) is longer than RA-DIT (Ground Truth). This observation further shows that the SFT training method may lead LLMs to overfitting the training signals, which affects the response length of LLMs.

