# OpenReview forum: "RAG-DDR: Optimizing Retrieval-Augmented Generation Using Differentiable Data Rewards"
_ICLR.cc/2025/Conference — ICLR 2025 Poster_

### Official Review · Reviewer_LZe5 · 2024-10-27

**Soundness:** 3
**Presentation:** 3
**Contribution:** 3
**Rating:** 6
**Confidence:** 4

**Summary:**

This paper introduces RAG-DDR (Retrieval-Augmented Generation with Differentiable Data Rewards), a novel method for optimizing RAG systems using reinforcement learning. Rather than using traditional supervised fine-tuning approaches, RAG-DDR uses a reward-based learning system to optimize both the knowledge retrieval and generation components. The method specifically focuses on aligning data preferences between different modules in the RAG pipeline to improve overall system performance, particularly in handling conflicts between parametric and external knowledge.

**Strengths:**

The approach is novel to RAG optimization. By using differentiable rewards, it effectively addresses the knowledge conflict problem that often plagues RAG systems.

**Weaknesses:**

1, The computational cost is high. RAG-DDR requires significantly more computational resources than traditional methods, requiring about 5x more API calls than standard approaches. This could limit its practical applicability in resource-constrained settings.
2, The paper also shows some limitations in complex reasoning tasks, particularly in multi-hop reasoning scenarios where the model can sometimes prematurely conclude questions are invalid. While the method shows strong performance on factual question-answering tasks, its effectiveness on more complex reasoning tasks could be improved.
3, while the paper demonstrates strong empirical results, it could benefit from a more thorough theoretical analysis of why the method works so well, particularly in terms of how it achieves better alignment between different RAG components.

**Questions:**

NA

---

> ### Author Response · Authors · 2024-11-22
>
> **Weaknesses:**
>
> **Reply to Weaknesses 1**: Our proposed DDR method does not increase the computational cost. Both training and inference are performed entirely with white-box models, eliminating any overhead from external API calls. Furthermore, our inference process remains identical to the standard RAG pipeline, without introducing extra computational resource demands. The primary computational cost arises from calculating the reward using the rollout method. However, model inference remains efficient, and the training stage is completed quickly (without training with large amounts of data).
>
>
> **Reply to Weaknesses 2**: For these multi-hop QA tasks, the one-time retrieval stage may limit the performance of RAG systems compared to the iterative reasoning methods. The reason may lie in that the one-time retriever may not supply sufficient information to answer the question [3]. Furthermore, RAG-DDR shows much better performance on ASQA, showing its effectiveness in training RAG systems to deal with complex reasoning tasks. The ASQA dataset asks models to explain the source of ambiguity in the question and connect all the valid short answers into a coherent passage, which also needs complex reasoning [2].
>
> |Model|2WikiMultihopQA [1] (accuracy)|ASQA [2] (rouge)|
> |-----|----|-----|
> | **MiniCPM-2.4B-SFT**  | | |
> |Vanilla RAG|22.6|9.9|
> |RA-DIT|25.9|13.2|
> |RAG-DDR|26.7|31.1|
>
> References:
>
> [1] Ho X, Nguyen A K D, Sugawara S, et al. Constructing A Multi-hop QA Dataset for Comprehensive Evaluation of Reasoning Steps. Proceedings of the 28th International Conference on Computational Linguistics.
>
> [2] Stelmakh I, Luan Y, Dhingra B, et al. ASQA: Factoid Questions Meet Long-Form Answers.  Proceedings of the 2022 Conference on Empirical Methods in Natural Language Processing. 2022: 8273-8288.
>
> [3] Trivedi H, Balasubramanian N, Khot T, et al. Interleaving Retrieval with Chain-of-Thought Reasoning for Knowledge-Intensive Multi-Step Questions. In Proceedings of ACL. 2023.
>
>
> **Reply to Weaknesses 3**: Thank you for your advice. In fact, we have conducted a comprehensive experimental analysis as shown in Figure 2, Table 3, Table 4, and Figure 6 to validate the effectiveness of our method. The experimental results reveal that DDR significantly enhances the utilization of external knowledge and mitigates the issue of knowledge conflict by aligning the data preferences across each module. We will add more theoretical analyses in the next version.

---

> ### Author Response · Authors · 2024-11-25
> **A gentle reminder**
>
> Dear Reviewer LZe5,
>
> Thank you for your time and efforts again in reviewing our paper. We would like to gently remind you that the discussion period is nearing its conclusion. In our submitted rebuttal, we have carefully addressed all your comments and suggestions, and we hope our responses have effectively clarified any concerns or questions you have raised.
>
> If there are any remaining points or if you require further clarification, please feel free to reach out. We sincerely appreciate your attention and valuable feedback.
>
> Best regards,
>
> Authors

---

> > ### Comment · Reviewer_LZe5 · 2024-12-03
> >
> > Thank you, I will keep my score

---

### Official Review · Reviewer_1cKK · 2024-11-02

**Soundness:** 3
**Presentation:** 3
**Contribution:** 2
**Rating:** 6
**Confidence:** 3

**Summary:**

The paper proposes the DDR framework for tuning agentic systems. The key idea is to sample multiple actions from an agent, and score them based on the downstream reward they fetch (computed via unrolling all subsequent agents in the chain). The actions are ordered by this score, and the DPO method is used to increase the likelihood of high scoring actions over low scoring actions.

The framework is applied to optimize a two agent RAG system consisting of a knowledge refinement agent (selecting relevant documents for RAG), and a generation agent that generates the response based on the selected document.

Through experiments and case studies, the authors show that the framework can perform better than several baselines, and can be better balance internal parametric knowledge with external information.

**Strengths:**

The paper addresses an important concern in RAG system of balancing use of external information versus internal parametric knowledge
of the generation module. This is common problem for production RAG systems where retrieval is never perfect and often suffers from conflicting and/or missing information.

The idea of jointly tuning retrieval and generation modules is interesting and not studied as much in the literature (although some prior art does exist; see next section).

The results in Sections 5.3 and 5.4 are compelling and show a nice advancement over baseline methods in dealing with noisy retrieval in RAG systems.

**Weaknesses:**

The main weakness of the paper is inadequate comparison with related approaches on jointly tuning retriever and generator modules, which make it hard to appreciate the technical contribution of the paper.

There is prior work on supervising the retrieval module of RAG system based on performance of the generator (e.g., Wang et al. 2023, https://arxiv.org/pdf/2311.08377). In my eyes, the novelty of this paper is the use of DPO instead of a supervised tuning (which would just maximize likelihood of action fetching highest reward). Thus, if I understand correctly, the novelty of this work is the negative gradient induced by the contrastive DPO loss. However, this is not discussed in the paper. It would help to study the impact of the negative gradient in depth, via ablation studies. For instance, is the negative gradient responsible for the claim in Section 5.3: "DDR can help learn more factual knowledge during training while also preventing the loss of previously memorized information".

The paper highlights RAG-DDR's ability to effectively combine parametric knowledge with external knowledge. To assess this, it would be helpful to see a comparison with adaptive approaches that use different retrieval modules (including no retrieval) for different queries. For instance, see: Jeong et al., 2024: https://arxiv.org/pdf/2403.14403

Other helpful ablations and baselines to experiment with:
* Section 3.2 notes that RAG-DDR tunes the generation module first followed by the knowledge retriever (KR) module. It would be helpful to perform ablations where the KR module is tuned before the generation module (as in Wang et al,. 2023).

* When evaluating vanilla LLM and vanilla RAG, it would help to experiment with large LLMs like Gemini-1.5-pro, GPT4o to understand if the inability to balance parametric knowledge vs. external knowledge is an artifact of smaller LLMs.

The paper introduces the framework of differentiable data rewards (DDR). However, the framework is only evaluated for simple two agent systems. Consequently, DDR does not come across as a fully developed contribution. Instead, the key contribution, at least to me, is the use of DPO in jointly trying retrieval and generation modules. I would encourage the authors to build on this aspect.

**Questions:**

Suggestion: I would encourage the authors to perform the ablation studies and related work comparisons suggested in the "Weaknesses" section.

Some questions about the KR module:
- The KR module generates binary Yes/No responses for the relevance of each document to the query. How is the binary output used to rank documents and identify the top-k documents?
- Line 216 says:  "document di that leads the agent system to achieve the highest evaluation reward is considered positive". Does this mean a single document is selected and fed to the generator module? But equation 7 shows that k documents are fed. Am I missing something?
- If it is a single document then when tuning the KR module do we consider the highest-rewards document as positive, and all others as negative?
- Would it make sense to consider contrastive objectives like ensuring P(Yes | q, d_i) > P(Yes | q, d_j) whenever the reward fetched by d_i is higher than d_j?

---

> ### Author Response · Authors · 2024-11-22
>
> **Weaknesses**
>
> **Reply to Weakness 1. The main weakness of the paper is inadequate comparison with related approaches on jointly tuning retriever and generator modules, which make it hard to appreciate the technical contribution of the paper.**:
>
> Thanks for your suggestion. We would like to take this opportunity to address and clarify some potential misunderstandings.
>
> a. Our main innovation is to propose an effective method for optimizing multi-agents within the RAG system, rather than merely substituting the SFT loss with the DPO loss. Furthermore, we have conducted experiments in Figure 2 and Figure 7 to demonstrate the effectiveness of our training strategy in helping learn factual knowledge while preventing the loss of previously memorized information.
>
> b. We compare RAG-DDR to the baseline model Filco [1] you mentioned. Please refer to the second point of our General Response. For the other baseline model, Adaptive-RAG [2], we are unable to provide its experimental results due to incomplete open source code.
>
> References:
>
> [1] Wang Z, Araki J, Jiang Z, et al. Learning to filter context for retrieval-augmented generation. 2023.
>
> [2] Jeong S, Baek J, Cho S, et al. Adaptive-RAG: Learning to Adapt Retrieval-Augmented Large Language Models through Question Complexity. In Proceedings of NAACL. 2024.
>
> **Reply to Weakness 2. Other helpful ablations and baselines to experiment with:**
>
> **Reply to Weakness 2.1:** We have already conducted these experiments, as shown in Table 6. The results indicate that the performance of RAG-DDR ($V\text{KR} First$) and RAG-DDR ($V\text{Gen} First$) is indistinguishable across different datasets and $V_\text{Gen}$, demonstrating the robustness of the DDR method to training order variations.
>
> **Reply to Weakness 2.2:** The challenge of balancing parametric and external knowledge is not unique to smaller LLMs but persists in larger models like ChatGPT. Studies [1] show that as models improve, they develop some awareness of knowledge conflicts and avoid blindly trusting external sources. However, larger LLMs, such as ChatGPT, often prioritize their incorrect internal knowledge over accurate external information, highlighting that this imbalance is a common issue across LLMs.
>
> Due to the high cost of closed-source models, we utilize the larger open-source model Qwen2.5-14B-Instruction as the generation module. As shown in the table below, our proposed DDR method remains effective.
> |Method|NQ|TriviaQA|MARCO QA|HotpotQA|T-REx|WoW|
> |-------|------|------|----------|---------|-------|------|
> |Vanilla RAG|48.1|81.1|18.5|29.4|37.3|14.3|
> |RAG-DDR|51.2|84.1|23.0|35.1|44.6|16.6|
>
> References:
>
> [1] Jin Z, Cao P, Chen Y, et al. Tug-of-war between knowledge: Exploring and resolving knowledge conflicts in retrieval-augmented language models. 2024.
>
> **Reply to Weakness 3. The paper introduces the framework of differentiable data rewards (DDR). However, the framework is only evaluated for simple two agent systems. Consequently, DDR does not come across as a fully developed contribution. Instead, the key contribution, at least to me, is the use of DPO in jointly trying retrieval and generation modules. I would encourage the authors to build on this aspect:**
>
> Thank you for your suggestion. We conduct these experiments, please refer to the first point of our General Response.

---

> > ### Comment · Reviewer_1cKK · 2024-11-28
> >
> > Thank you for the additional experiments on comparing against additional baselines and demonstrating that the method can work well with multiple agents.
> >
> > I see that RAG-DDR is superior to Filco but I am still struggling to understand the source of this gain (beyond my previous speculation on the use of DPO loss). If we compare Filco with the "Conditional Cross-Mutual Information" context selector to RAG-DRR where we train the KR module first then the only major difference seems to be the SFT loss and DPO loss.
> >
> > Given the additional work put in I am willing to increase my rating to 6.
> >
> > However, I would strongly encourage the authors to shed light on why the method works well compared to baselines like Filco.
> > (To me concept clarity on why a method works is just as important as empirical superiority.)

---

> ### Author Response · Authors · 2024-11-22
>
> **Questions:**
>
> **Reply to Question 1:** In designing the KR module, we adopt the implementation of kotaemon (https://github.com/Cinnamon/kotaemon/blob/main/libs/kotaemon/kotaemon/indices/rankings/llm.py), and the prompts used for KR are provided in Figure 5.
>
> Specifically, the KR module processes the $n$ retrieved documents in descending order of retrieval scores to assess their usefulness for answering the question. For each document, the module outputs “Yes” to retain it if deemed helpful or “No” to discard it. This continues until $k$ documents are retained, ensuring the top-$k$ most relevant documents are selected.
>
> **Reply to Question 2:** During the training of the Knowledge Refinement module, we individually feed each document to the Generation Model to obtain the reward used for training the Knowledge Refinement module (line 216). This is because the Knowledge Refinement module determines whether a query is relevant to an individual document and generates Yes/No labels for each query-document pair. Equation 7 shows that the Generation Model is fed $k$ documents for sampling, which serves the same purpose as the function of the generation module—feeding $k$ documents and answering the query.
>
> **Reply to Question 3:** During training of the Knowledge Refinement module, we randomly select one document with the highest reward score and one with the lowest reward score from the document collection to create preference data pairs for training the entire RAG system.
>
> **Reply to Question 4:** Thank you for your suggestion and we will explore it in the future. Our main contribution lies in proposing a method to optimize the multi-agent of the RAG system. Therefore, we adhere to standard practices for module selection and design the KR module by following the implementations of kotaemon(https://github.com/Cinnamon/kotaemon/blob/main/libs/kotaemon/kotaemon/indices/rankings/llm.py).

---

> ### Author Response · Authors · 2024-11-25
> **A gentle reminder**
>
> Dear Reviewer 1cKK,
>
> Thank you for your time and efforts again in reviewing our paper. We would like to gently remind you that the discussion period is nearing its conclusion. In our submitted rebuttal, we have carefully addressed all your comments and suggestions, and we hope our responses have effectively clarified any concerns or questions you have raised.
>
> If there are any remaining points or if you require further clarification, please feel free to reach out. We sincerely appreciate your attention and valuable feedback.
>
> Best regards,
>
> Authors

---

> ### Author Response · Authors · 2024-11-28
>
> Thank you once again for your insightful review. Your comments provide valuable feedback and offer us an opportunity to improve our work.
>
> We will clarify why the DDR approach outperforms baselines like Filco. Filco optimizes the context selector and generation model independently using SFT. While Falco’s context selector effectively filters out irrelevant information, the retained content may not always benefit the generation model. This is because the context selector in Filco is trained using labeled data rather than feedback from the downstream generation model, causing its outputs to align more with the training data than with the knowledge needs of the generation model. As highlighted by works like AAR [1] and REPLUG [2], incorporating feedback signals from the generation model to train the retrieval model can better align them, ultimately providing higher-quality documents for generation.
>
> References
>
> [1] Yu Z, Xiong C, Yu S, et al. Augmentation-Adapted Retriever Improves Generalization of Language Models as Generic Plug-In. Proceedings of the 61st Annual Meeting of the Association for Computational Linguistics. 2023.
>
> [2] Shi W, Min S, Yasunaga M, et al. REPLUG: Retrieval-Augmented Black-Box Language Models. Proceedings of the 2024 Conference of the North American Chapter of the Association for Computational Linguistics: Human Language Technologies (Volume 1: Long Papers). 2024.

---

> > ### Comment · Reviewer_1cKK · 2024-11-29
> >
> > I see your point. However, if I understand correctly, your system reward is also based on matching with the final response. (I gathered this from line 289 which says that reward is based on rouge-l and accuracy.)
> >
> > In this sense the differences with Filco come down to:
> > - fuzzy match instead of exact matching with supervised response
> > - dpo loss vs sft loss
> >
> > In my opinion, telling these two apart will greatly enhance conceptual understanding of the method.

---

> > > ### Author Response · Authors · 2024-12-01
> > >
> > > Thank you very much for your response to our reply! As you mentioned, the difference between Filco and RAG-DDR lies in the source of the supervised training signals and the difference in the training loss function. Emphasizing this distinction helps to understand the DDR method better.
> > >
> > > Furthermore, another key difference between the DDR method and Filco is that DDR is not limited to training the text filtering and generation modules; it can also be extended to RAG systems with more modules.

---

### Official Review · Reviewer_QfBP · 2024-11-04

**Soundness:** 3
**Presentation:** 3
**Contribution:** 3
**Rating:** 6
**Confidence:** 4

**Summary:**

This paper introduces Differentiable Data Rewards (DDR), a novel method for training Retrieval-Augmented Generation (RAG) systems. DDR collects rewards via a rollout method to optimize each agent while aligning data preferences between different agents within the system. The method addresses two key challenges in RAG systems: (1) training signal overfitting and (2) poor robustness against conflicting knowledge.

**Strengths:**

- Overall, the presentation is good and the method is easy-to-follow.
- The paper introduces the idea of DDR, which enables end-to-end optimization of RAG systems. The proposed method of aligning data preferences between knowledge refinement and generation agents alleviates the robustness issue when encountering external knowledge conflicts.
- The experimental results show that RAG systems trained with DDR strategy demonstrate improvements in terms of knowledge usage and robustness against knowledge conflicts.

**Weaknesses:**

- Figure 1 is not clear enough. It could be better to illustrate the correspondence between the left and right parts, as well as the order of two modules in the right part (could use step 1, 2 as annotations). Additionally, it would be helpful to indicate how the reward signal propagates throughout the entire training process.


- I realize that training large language models is resource-intensive, but the demonstrations in the paper could be more convincing by including:

    - (If possible)Additional experimental results from larger-scale LLMs, such as Qwen2.5-14B or other larger models.
    - (If possible)Additional evaluations using more different LLM combinations to compose the RAG system (It seems that the current setup consider Llama3-8B-Instruct as knowledge refinement module and MiniCPM-2.4B/Llama3-8B-Instruct as generator)

**Questions:**

1. Kindly refer to the Weakness section.

2. Could the authors define what is meant by “data preferences between different RAG modules”? This concept appears multiple times in the paper, yet it remains somewhat unclear to me. Given that RAG primarily addresses knowledge-based tasks, could “preferences” here be more aligned with factual accuracy? I would appreciate a brief definition, ideally illustrated with a case study that demonstrates why this issue is important. It would be especially helpful if different base models were involved, as their internal knowledge may vary, potentially influencing their data preferences.

3. In addition, regarding the rollout method:

- What are the training speed and computational requirements? How does the method scale with larger datasets in terms of efficiency?
- Could the DDR method be expanded beyond just two agents? A more in-depth discussion from the authors on this point would offer valuable insights for developing more complex RAG systems.

4. An interesting finding in Appendix A.4 is that DDR enhances MiniCPM-2.4B’s mathematical capabilities. Could the authors provide a brief analysis explaining the possible reasons for this?

5. Typo: Line 269, should be "Llama3-8B-Instruct".

---

> ### Author Response · Authors · 2024-11-22
>
> **Weaknesses:**
>
> **Reply to Weakness 1**: Thank you for your suggestion. We will revise and update Figure 1 to address these issues in the next version of the paper.
>
> **Reply to Weakness 2**:
>
> **Reply to Weakness 2.1**: Thank you for your valuable suggestion. In response, we have conducted experiments with large-scale LLMs. Specifically, we utilized Qwen2.5-14B-Instruct as the generation module and Llama3-8B-Instruct as the refinement module to build a new combination for the RAG system.
> |Method|NQ|TriviaQA|MARCO QA|HotpotQA|T-REx|WoW|
> |-|-|-|-|-|-|-|
> | **Qwen2.5-14B-Instruction**  | | | | | | |
> |Vanilla RAG|48.1|81.1|18.5|29.4|37.3|14.3|
> |RAG-DDR |51.2|84.1|23.0|35.1|44.6|16.6|
>
> **Reply to Weakness 2.2**: We have implemented the RAG model using the following combinations: (Llama3-8B-Instruct + MiniCPM-2.4B-SFT), (Llama3-8B-Instruct + Llama3-8B-Instruct), and (Llama3-8B-Instruct + Qwen2.5-14B-Instruction). In future work, we aim to explore additional model configurations to further enhance performance.
>
>
> **Questions:**
>
> **Reply to Question 1**:  Please refer to the weakness section.
>
> **Reply to Question 2**: Data preferences refer to the specific input and output data that each module in a RAG system depends on, enabling the entire system to generate more accurate responses. Specifically, each module processes input data and generates output data, which are then passed to subsequent modules. The preferred data for each module are those inputs and outputs that contribute to producing correct answers across the entire RAG system. The DDR training objective seeks to align the input and output data of each module with these preferred data, thereby optimizing the overall performance of the RAG system.
>
> **Reply to Question 3:**
>
> **Reply to Question 3.1:** In our experiments, the DDR method consists of two stages: data forward propagation and differentiable data reward. These stages aim to construct the training data and train the RAG module, respectively. For example, during training the generation module (Llama3-8B-Instruct), data forward propagation takes approximately 35 hours, while differentiable data reward requires around 1 hour and 40 minutes on an 8-GPU A100 server. During training the knowledge module (Llama3-8B-Instruct), data forward propagation takes about 12 hours, with differentiable data reward taking approximately 1 hour and 50 minutes on the same setup.
>
> Although the data forward propagation process is time-intensive for generating training data, the actual model training is relatively quick. Moreover, DDR method does not require a large amount of training data and training time. As shown in the table below, using the training data described in this paper, Llama3-8B-Instruct and MiniCPM-2.4B-SFT achieve convergence of dev loss with less than one epoch's worth of training data.
>
> |Model|800|1600|3200|8000|16000|16800|17600|
> |-----|----|-----|----|-----|----|-----|----|
> |Llama3-8B-Instruct|0.637|0.599|0.586|0.519|0.488|0.491|0.490|
>
> |Model|800|1600|3200|8000|19200|20000|20800|
> |-----|----|-----|----|-----|----|-----|----|
> |Minicpm-2.4B-SFT|0.574|0.549|0.533|0.474|0.434|0.436|0.434|
>
> **Reply to Question 3.2:** Thank you for your suggestion. We conduct these experiments, please refer to the first point of our General Response.
>
> **Reply to Question 4**: MiniCPM-2.4B-SFT is a smaller parameter model with weaker mathematical reasoning capabilities compared to larger models. We believe its improved performance may be attributed to the inclusion of Math QA and reasoning tasks in the collected training data.
>
> **Reply to Question 5**: Thank you for your suggestion. We will address this spelling error in the next version of the paper.

---

> ### Author Response · Authors · 2024-11-25
> **A gentle reminder**
>
> Dear Reviewer QfBP,
>
> Thank you for your time and efforts again in reviewing our paper. We would like to gently remind you that the discussion period is nearing its conclusion. In our submitted rebuttal, we have carefully addressed all your comments and suggestions, and we hope our responses have effectively clarified any concerns or questions you have raised.
>
> If there are any remaining points or if you require further clarification, please feel free to reach out. We sincerely appreciate your attention and valuable feedback.
>
> Best regards,
>
> Authors

---

> ### Comment · Reviewer_QfBP · 2024-11-28
>
> Thank you for your reply. I have read your reply, I believe most of my concerns have been addressed, and I will raise my score.
>
> Additionally, when comparing Qwen2.5-14B with the other models in Table 1, I noticed that Qwen2.5-14B performs significantly better than LLaMA3-8B in the vanilla RAG setting. However, after training, their performances are comparable. Based on my experience, I personally believe this discrepancy is due to the distinct “tunability” of the base models (e.g., differences in how they learn from fine-tuning data, catastrophic forgetting, etc.). Thus, this is a minor concern on my part.
>
> However, going further and considering your General Response 1, I am curious about the major bottlenecks in the framework that limit its performance. I understand that exploring this experimentally involves significant costs, but I hope the authors could provide some discussion on this topic and potentially explore it further in the future, if feasible.

---

> ### Author Response · Authors · 2024-11-28
>
> Thank you once again for your insightful review. Your comments provide valuable feedback and offer us an opportunity to improve our work.
>
> * As you mentioned, Qwen2.5-14B outperforms LLaMA3-8B in the vanilla RAG setting for tasks like NQ but becomes comparable to LLaMA3-8B after training. This discrepancy may stem from inherent differences between the base models. Furthermore, the preference data generated during DPO training can vary significantly across models, potentially affecting generation results. We plan to explore this further by investigating a wider range of model combinations in future work.
>
> * The primary bottleneck of the DDR approach lies in the time cost of generating training data. To maximize the training effect of DDR, we use a rollout method to compute reward scores for each module, ensuring they align with the RAG system’s final output. While this increases the data construction time, the process is conducted offline and does not affect training efficiency.

---

### Official Review · Reviewer_oqk9 · 2024-11-04

**Soundness:** 3
**Presentation:** 3
**Contribution:** 3
**Rating:** 6
**Confidence:** 3

**Summary:**

This paper introduces RAG-DDR, a novel reinforcement learning-based approach designed to improve retrieval-augmented generation (RAG) systems. The authors conceptualize the RAG framework as comprising two key modules: a knowledge refinement module, $V_{\text{KR}}$​, and a generation module, $V_{\text{Gen}}$. RAG-DDR computes partial reward for each module's response with a rollout process. Preference pairs are then constructed based on the rewards and then being used to optimize $V_{\text{Gen}}$ and $V_{\text{KR}}$ through DPO. Specifically, for $V_{\text{Gen}}$ (with filtered documents), the response with highest score is considered positive and lowest score is considered negative. For $V_{\text{KR}}$, the document that, when passed through $V_{\text{Gen}}$, yields the highest response score is considered positive, while the document with the lowest score is negative.

The authors perform extensive experiments across various knowledge-intensive tasks to evaluate RAG-DDR, comparing it against the vanilla RAG and SFT-based methods, showing that RAG-DDR can generate more accurate response and has the ability to leverage LLM's internal knowledge while not being misled by the external noisy information. Ablation studies further demonstrate the effectiveness of optimizing both $V_{\text{KR}}$ and $V_{\text{Gen}}$ using the proposed DDR approach.

**Strengths:**

1. Extensive experiments together with detailed analysis. In particular, the result from Table 3 is interesting, as it shows RAG-DDR's ability to balance between using internal and external knowledge.
2. The overall methodology is straightforward yet effective, leveraging a DPO-based approach to facilitate self-training.

**Weaknesses:**

1. Some aspects are not clear, see the question section.
2. (Minor thing) Typo at line 850 "identify the positive and positive responses for each query." -> "identify the positive and negative responses for each query."

**Questions:**

1. What distinguishes the proposed DDR method from step-DPO? Is that the only difference comes from the base policy, i.e., in DDR the base policy at each step comes from different agents while for step-DPO the base policy at each step are shared?
2. What is the impact of training RAG-DDR iteratively over multiple rounds? While results for 2-round training are reported in Table 1, a more detailed analysis of the outcomes would be beneficial.
3. In section 5.3, the authors argue that "SFT training method tends to cause LLMs to overfit the supervised data" from the observation that RA-DIT's generated responses is significantly shorter than vanilla RAG and RAG-DDR's responses. While I generally agree with this statement, I wonder whether this phenomenon could be attributed to differences in the average response length within the training sets for RA-DIT and RAG-DDR. If so, what would occur if we trained RA-DIT using the positive samples from RAG-DDR? Would this adjustment also impact the model's accuracy?

---

> ### Author Response · Authors · 2024-11-22
>
> **Weaknesses:**
>
> **Reply to Weakness 1**:  Please refer to the Question section.
>
> **Reply to Weakness 2**: Thank you for your comments and suggestions. We will carefully proofread our paper.
>
> **Questions:**
>
> **Reply to Question 1**: Thank you for your suggestion. We have clarified the differences between STEP-DPO [1] and DDR in the final paragraph of the Related Work section and will further highlight this distinction in the next version of our paper.
>
> Specifically, STEP-DPO focuses on optimizing a single agent by splitting the model’s responses into multiple steps and evaluating the correctness of each step individually. In contrast, DDR is tailored for optimizing a multi-agent system, employing a rollout mechanism to gather system-level rewards for each agent and subsequently refining the agents based on these rewards.
>
> References:
>
> [1] Lai X, Tian Z, Chen Y, et al. Step-dpo: Step-wise preference optimization for long-chain reasoning of llms. 2024.
>
>
> **Reply to Question 2**: Thank you for your insightful suggestion. To further examine the effectiveness of multi-round optimization, we conducted an analysis of RAG-DDR (2-Round) across the same three scenarios presented in Table 3 of the paper. As illustrated in the table below, RAG-DDR (2-Round) consistently outperforms RAG-DDR (1-Round) across all three scenarios, with particularly significant gains observed in the Internal Knowledge scenario. These findings indicate that 2-round DDR training can further improve the model’s ability to balance internal and external knowledge, especially for larger models.
>
> | Method  | Has-Answer (NQ) | Has-Answer (HotpotQA) | Has-Answer (T-REx) | Miss-Answer (NQ) | Miss-Answer (HotpotQA) | Miss-Answer (T-REx) | Internal Knowledge (NQ) | Internal Knowledge (HotpotQA) | Internal Knowledge (T-REx) |
> |--|--|--|--|--|--|--|--|--|--|
> | **MiniCPM-2.4B-SFT**     | || |  |  |  |  |   |   |
> | RAG-DDR (w/ 1-Round) | 65.5 | 56.9    | 52.6  | 2.4| 10.8  | 5.9   | 82.9   | 81.0 | 78.5  |
> | RAG-DDR (w/ 2-Round) | 65.3  | 59.4  | 51.7  | 2.4   | 14.2   | 5.5  | 84.0  | 82.3 | 74.8   |
> | **Llama3-8B-Instruct** | || |  |  |  |  |   |   |
> | RAG-DDR (w/ 1-Round) | 69.5 | 64.3 | 59.9 | 4.1  | 18.0|  5.4    | 88.4  | 82.0  | 79.2   |
> | RAG-DDR (w/ 2-Round) | 71.6   | 66.2  | 65.4   | 5.6 | 18.9  | 7.7   | 89.3 | 83.9 | 89.7   |
>
>
> **Reply to Question 3**: Thank you for your valuable suggestion. We use positive samples from RAG-DDR to train RA-DIT. Then, we evaluate the downstream performance of RA-DIT and analyze its response length. The corresponding results are presented in the tables below.
>
> a. The model performance is shown in the below table. RAG-DDR consistently outperforms RA-DIT across training setups, whether using ground-truth or DPO-positive samples, highlighting the effectiveness of our DDR training method. Furthermore, RA-DIT trained on DPO-positive samples performs worse than when trained on ground-truth data, suggesting that aligning solely with noisy positive examples is insufficient without the contrast provided by negatives. This emphasizes the advantage of our optimization approach leveraging the DPO loss.
>
>
> | Model | NQ   | TQA  | MARCOQA | HotpotQA | T-REx | WoW  |
> |--|--|--|--|--|--|--|
> | **MiniCPM-2.4B-SFT** |||||||
> | RA-DIT (Ground Truth) | 41.8 | 78.6 | 19.6 | 26.1| 25.2  | 15.7 |
> | RA-DIT (Positive Label) | 40.9 | 77.2 | 19.9 | 22.0| 25.3  | 14.7 |
> | RAD-DDR  | 47.0 | 82.7 | 28.1 | 32.5 |32.6  | 17.2 |
> | **Llama3-8B-Instruct**  |||||||
> | RA-DIT (Ground Truth) | 46.2 | 87.4 | 20.3    | 34.9     | 41.7  | 14.8 |
> | RA-DIT (Positive Label) | 45.8 | 86.4 | 23.0    | 31.4     | 25.3  | 13.2 |
> | RAD-DDR| 50.7 | 88.2 | 25.1    | 37.3     | 37.3  | 14.9 |
>
> b. The response lengths are presented in the table below. Notably, the response length of RA-DIT trained with DPO-positive samples is longer than that of RA-DIT trained with ground truth. This observation further shows that the SFT training method may lead LLMs to overfit the supervised data, thereby affecting the model's response length.
>
>
> | Model   | NQ   | TQA  | MARCOQA | HotpotQA | T-REx | WoW   |
> |--|--|--|--|--|--|--|
> | **MiniCPM-2.4B-SFT** |||||||
> | Vanilla RAG        | 12.5 | 4.7  | 32.4    | 5.6      | 4.4   | 48.5  |
> | RA-DIT (Ground Truth) | 7.6  | 4.8  | 13.8    | 8.9      | 5.0   | 17.2  |
> | RA-DIT (Positive Samples) | 19.2 | 8.3  | 34.6    | 13.7     | 6.7   | 32.8  |
> | RAD-DDR            | 27.4 | 16.9 | 47.8    | 27.2     | 11.2  | 56.6  |
> | **Llama3-8B-Instruct** |||||||
> | Vanilla RAG | 49.6 | 25.0 | 85.4    | 43.9     | 34.4  | 98.5  |
> | RA-DIT (Ground Truth) | 6.9  | 8.7  | 14.5    | 23.6     | 4.0   | 16.7  |
> | RA-DIT (Positive Samples) | 42.1 | 14.8 | 72.2    | 24.0     | 3.9   | 72.2  |
> | RAD-DDR | 56.0 | 33.9 | 81.2    | 51.6     | 35.2  | 101.5 |

---

> ### Author Response · Authors · 2024-11-25
> **A gentle reminder**
>
> Dear Reviewer oqk9,
>
> Thank you for your time and efforts again in reviewing our paper. We would like to gently remind you that the discussion period is nearing its conclusion. In our submitted rebuttal, we have carefully addressed all your comments and suggestions, and we hope our responses have effectively clarified any concerns or questions you have raised.
>
> If there are any remaining points or if you require further clarification, please feel free to reach out. We sincerely appreciate your attention and valuable feedback.
>
> Best regards,
>
> Authors

---

### Official Review · Reviewer_FTLt · 2024-11-05

**Soundness:** 4
**Presentation:** 3
**Contribution:** 2
**Rating:** 6
**Confidence:** 4

**Summary:**

This paper focuses on improving the retrieval-augmented generation (RAG) system by fine-tuning the content filter and the LLM generator while keeping the retriever frozen. It claims that existing supervised fine-tuning (SFT) approaches fail to model data preferences for different modules in the RAG system (i.e., the filter and generator) and tend to overfit training signals. In this work, the authors introduce a training method called differentiable data rewards (DDR), which uses DPO loss to train both the filter and generator. Experiments on six knowledge-intensive tasks show superior performance compared to several off-the-shelf RAG baselines and a fine-tuned RAG baseline (though without the filter module to denoise the retrieved contents).

**Strengths:**

=== Strengths ===

S1. The motivation of this work is clear, and the paper is easy to follow.

S2. The idea of introducing DPO loss to train the content filter and LLM generator in RAG is novel (although its significance is not quite clear given the current empirical study, see Weakness 1).

S3. The main comparison between the proposed method and baseline methods, along with the ablation study, is evaluated on a wide range of knowledge-intensive tasks.

**Weaknesses:**

=== Weaknesses ===

W1. Potentially problematic experimental setting. The main claimed contribution is proposing a DPO loss to train the content filter and LLM generator. However, in the experiments, the proposed method is primarily compared to off-the-shelf RAG methods without any training. The only fine-tuned baseline is RA-DIT, which only trains the LLM generator. Does RA-DIT also leverage the content filter module during evaluation? If not, it is unclear whether the improved evaluation performance is due to filtering or to the better optimization by DPO loss compared to traditional SFT loss. Furthermore, is “RAG w/ $V_{KR}$” the only RAG baseline that includes a content filter (i.e., knowledge refinement) module? The comparison in the main table and ablation study may be unfair and the results can be misleading if the proposed RAG-DDR can benefit from an additional filtering mechanism that other baselines lack of.

W2. Missing comparisons and discussion with closely relevant RAG baselines on denoising retrieved content. Approaches for mitigating conflicts between external retrieved knowledge and the LLM’s internal parametric knowledge, as well as for denoising retrieved content, have been widely explored in RAG literature [1,2,3,4]. A more thorough discussion of these works is needed to better situate this work. For instance, [2] directly trains the LLM generator using SFT loss with noisy inputs to improve robustness to noise without any filtering/knowledge refinement, [3] employs self-synthetic rationales to denoise retrieved content and guide generation for better accuracy, which is also trained with SFT loss, and [4] introduces an inference-time algorithm to deal with irrelevant retrieved documents through a mechanism called isolation-then-aggregation. Without a detailed comparison with such related works, it is difficult to assess the empirical significance of this work, especially since some of these works are strong RAG methods trained with SFT loss (e.g., [2,3]) for specific denoising purposes.

**References**

[1]. AstuteRAG: Overcoming Imperfect Retrieval Augmentation and Knowledge Conflicts for Large Language Models. arXiv 2024.

[2]. Making Retrieval-Augmented Language Models Robust to Irrelevant Context. ICLR 2024.

[3]. InstructRAG: Instructing Retrieval-Augmented Generation via Self-Synthesized Rationales. arXiv 2024.

[4]. Certifiably Robust RAG against Retrieval Corruption. arXiv 2024.

**Questions:**

Please address the questions in Weaknesses.

---

> ### Author Response · Authors · 2024-11-22
>
> **Reply to Weakness 1**: Thank you for your comments. We would like to take this opportunity to address and clarify some potential misunderstandings.
>
> a. In our experiments, all models are implemented using the same RAG modules, including the Knowledge Refinement module and Generation Module. RA-DIT also employs the Knowledge Refinement module ($V_\text{KR}$) during evaluation. Thus, we conduct a fair comparison.
>
> b. The notation in Table 2 may cause some misunderstanding; we will revise our paper to ensure all details are presented clearly. In fact, the model performance of the model $w/ V_\text{KR}$ in Table 2 is identical to that of the Vanilla RAG model in Table 1.
>
> **Reply to Weakness 2**: Thanks for your suggestion. We have included additional baselines for comparison, as detailed in the second point of our General Response.

---

> ### Author Response · Authors · 2024-11-25
> **A gentle reminder**
>
> Dear Reviewer FTLt,
>
> Thank you for your time and efforts again in reviewing our paper. We would like to gently remind you that the discussion period is nearing its conclusion. In our submitted rebuttal, we have carefully addressed all your comments and suggestions, and we hope our responses have effectively clarified any concerns or questions you have raised.
>
> If there are any remaining points or if you require further clarification, please feel free to reach out. We sincerely appreciate your attention and valuable feedback.
>
> Best regards,
>
> Authors

---

> ### Comment · Reviewer_FTLt · 2024-11-28
>
> I appreciate the author's efforts in providing detailed clarifications and new experimental results (though Yoran et al. (ICLR 2024) is still missing in the comparison) as additional support, which addressed most of my concerns. To reflect this, I will raise my score.
>
> However, I still believe the positioning of the paper could be improved, since some closely relevant works are not sufficiently discussed in the original manuscript as mentioned earlier. For example, Yoran et al. (ICLR 2024) was published **before July 1, 2024**, and their code is even [publicly available ](https://github.com/oriyor/ret-robust). Unfortunately, there is no comparison or discussion of this work in the paper.
>
> It's good to see the new results and to know the author's plan for revising the manuscript. Please ensure the updated version incorporates these discussions and results to better position and justify the contribution of this paper.

---

> ### Author Response · Authors · 2024-11-29
>
> Thank you once again for your insightful review. Your comments provide valuable feedback and offer us an opportunity to improve our work.
>
> For the missing baseline you mentioned,  we attempted to reproduce it.  Unfortunately, our reproduction was hampered by the lack of some experimental details in the open source code. Other researchers have encountered problems similar to ours [1]. We are trying to solve these problems and reproduce the baseline. For the updated experimental results in the rebuttal, we will add them to subsequent versions of the paper.
>
> References:
>
> [1] https://github.com/oriyor/ret-robust/issues/5

---

### Author Response · Authors · 2024-11-22
**General Response 2**

**More Baseline Models**

DDR focuses on optimizing RAG systems rather than denoising and filtering. Existing evaluation results have demonstrated the effectiveness of RAG-DDR methods, and our approach can also be applied to optimize RAG systems with varying numbers of agents.

To make our paper more complete, we have added the following experiments to demonstrate the effectiveness of RAG-DDR. However, some baselines could not be implemented due to two main reasons: first, certain referenced works are concurrent studies, and second, some are not open-source. We found that AstuteRAG [4], InstructRAG [1], and Certifiably Robust RAG against Retrieval Corruption [5] are **ICLR 2025 Submissions**. According to the review policy (https://iclr.cc/Conferences/2025/ReviewerGuide), these papers do not force comparisons.

a. InstructRAG [1]: InstructRAG sets the maximum generation length to 4096 due to its typically longer responses. For a fair comparison, we compare InstructRAG and RAG-DDR by setting the maximum lengths as 4096 and 32, which keep the settings of InstructRAG and RAG-DDR.

|Method|NQ|TriviaQA|HotpotQA|T-REx|
|-------|------|------|----------|---------|
|InstructRAG(max_len=32)|4.2|21.1|11.0|17.1|
|RAG-DDR(max_len=32)|50.7|88.2|37.3|37.3|
|InstructRAG(max_len=4096)|60.8|92.0|42.3|43.1|
|RAG-DDR(max_len=4096)|58.0|91.9|43.9|45.6|

When the generation length is set to 4096, RAG-DDR demonstrates comparable performance to InstructRAG on the NQ and TQA tasks, as InstructRAG is trained specifically on these datasets. However, for out-of-domain tasks, InstructRAG underperforms relative to RAG-DDR. When the maximum generation length is reduced to 32, RAG-DDR outperforms InstructRAG, indicating that RAG-DDR requires a shorter generation length to generate accurate answers, compared to InstructRAG.

b. Filco [2]: We train Flan-T5-XL as the context filtering model and Llama3-8B as the generation model using the code and the experimental settings provided by Filco. We compare the Filco with the RAG-DDR using the Llama3-8B as the generation module.
| Model  | NQ   | TQA  | MARCOQA | HotpotQA | T-REx | WoW   |
|--------------------|------|------|---------|----------|-------|-------|
|Filco  | 46.6| 81.9 | 14.6| 25.3|44.1|14.4|
| RAD-DDR   | 50.7 | 88.2 | 25.1  | 37.3 | 37.3  | 14.9 |

c. Self-RAG [3]: In addition to the mentioned baselines, we also include the Self-RAG model as an additional baseline. For Self-RAG, we use MiniCPM-2.4B-SFT and Llama3-8B-Instruct as the backbone models, training them with the open-source dataset provided for Self-RAG. Our results show that RAD-DDR outperforms Self-RAG across various backbone models and tasks.

|Method|NQ|TriviaQA|MARCO QA|HotpotQA|T-REx|WoW|
|---|---|---|---|---|---|---|
|MiniCPM-2.4B-SFT||||||
|Self-RAG|28.8|48.6|18.2|15.2|23.9|13.4|
|RAD-DDR|47.0|82.7|28.1|32.5|32.6|17.2|
|Llama3-8B-Instruct||||||
|Self-RAG|39.6|78.2|24.3|12.5|29.5|14.2|
|RAD-DDR|50.7|88.2|25.1|37.3|37.3|14.9|


References:

[1] Wei Z, Chen W L, Meng Y. InstructRAG: Instructing Retrieval-Augmented Generation with Explicit Denoising. 2024.

[2] Wang Z, Araki J, Jiang Z, et al. Learning to filter context for retrieval-augmented generation. 2023.

[3] Asai A, Wu Z, Wang Y, et al. Self-rag: Learning to retrieve, generate, and critique through self-reflection. ICLR 2024.

[4] Wang F, Wan X, Sun R, et al. Astute RAG: Overcoming Imperfect Retrieval Augmentation and Knowledge Conflicts for Large Language Models. 2024.

[5] Xiang C, Wu T, Zhong Z, et al. Certifiably Robust RAG against Retrieval Corruption. 2024.

---

### Author Response · Authors · 2024-11-22
**General Response 1**

We thank all reviewers for their efforts and valuable suggestions. Here we try to address the common concerns.

**Expanding the DDR method to more agents.**

In this experiment, we extend the DDR method to more agents to investigate its effectiveness in optimizing more complex RAG systems. Specifically, we utilize the Knowledge Refinement module, Summarization Module and Generation Module to build a new RAG pipeline:

* Knowledge Refinement module: it filters the retrieved documents as described in our paper.
* Summarization module: it summarizes the refined documents from the Knowledge Refinement module into concise summaries containing the essential knowledge needed to answer the questions.
* Generation module: it uses the summaries generated by the Summarization module as context to answer the questions.

In our experiments, we utilize Llama3-8B-Instruct as the backbone model for constructing the Knowledge Refinement and Summarization modules, while MiniCPM-2.4B-SFT serves as the backbone for the Generation module. The results of the RAG-DDR system with the additional Summarization module are shown in the table below. Vanilla RAG indicates that all three modules in the RAG system are not finetuned. RAG-DDR (Only Generation Module) indicates that only the Generation module is finetuned. RAG-DDR (Generation & Summarization Module) indicates that both the Generation and Summarization modules are finetuned. Finally, RAG-DDR (All) indicates that all three modules are optimized. The experimental results demonstrate that our DDR method remains effective even in RAG systems with more agents.


|Method|NQ|TriviaQA|MARCO QA|HotpotQA|T-REx|WoW|
|-------|------|------|----------|---------|-------|------|
|Vanilla RAG (Three Modules)|44.1|82.3|17.5|28.9|25.4|15.4|
|RAG-DDR (Only Generation Module) |46.6|83.6|26.9|31.2|28.3|16.9|
|RAG-DDR (Generation & Summarization Module) |47.6|83.9|26.4|	32.0|28.6|17.0|
|RAG-DDR (All)|47.9|84.8|26.4|32.9|28.7|17.3|

---

### Meta-Review · Area_Chair_Au8M · 2024-12-17

**Metareview:**

This paper introduces a new method for end-to-end training of Retrieval-Augmented Generation (RAG) systems using a technique with DPO loss. The authors train both a generator and a filter component, allowing the system to effectively identify and utilize relevant knowledge for generating accurate answers. Extensive experiments demonstrate the superiority of this approach compared to existing alternatives.

Strengths:
- The paper is well-written and the empirical results are comprehensive and convincing.
- The authors address a significant limitation in current RAG systems.

Weaknesses:
- The paper's positioning within the rapidly evolving landscape of end-to-end RAG tuning research is not entirely clear.
- The computational complexity of the training process could be a potential drawback.

Despite some weaknesses, the novel methodology, compelling empirical results, and the paper's focus on an important problem in RAG systems lead me to recommend acceptance.

**Additional Comments On Reviewer Discussion:**

The authors have diligently addressed many of the reviewers' concerns by providing further experiments demonstrating the generalizability of their method across different LLMs and tasks. They have also clarified the novelty of their approach and included comparisons with additional baselines. To further improve the manuscript, I recommend focusing on the remaining reviewer comments, particularly on better positioning the paper within the broader research context.

---

### Decision · Program_Chairs · 2025-01-22

Accept (Poster)